# Dependence of the Structural and Magnetic Properties on the Growth Sequence in Heterostructures Designed by YbFeO_3_ and BaFe_12_O_19_

**DOI:** 10.3390/nano14080711

**Published:** 2024-04-18

**Authors:** Sondes Bauer, Berkin Nergis, Xiaowei Jin, Reinhard Schneider, Di Wang, Christian Kübel, Petr Machovec, Lukas Horak, Vaclav Holy, Klaus Seemann, Tilo Baumbach, Sven Ulrich

**Affiliations:** 1Institute for Photon Science and Synchrotron Radiation, Karlsruhe Institute of Technology, Hermann-von-Helmholtz-Platz 1, D-76344 Eggenstein-Leopoldshafen, Germany; berkin.nergis@kit.edu (B.N.); tilo.baumbach@kit.edu (T.B.); 2Laboratory for Electron Microscopy, Karlsruhe Institute of Technology, Engesserstr. 7, D-76131 Karlsruhe, Germany; carriorjin@gmail.com (X.J.); reinhard.schneider@kit.edu (R.S.); 3Institute of Nanotechnology, Karlsruhe Institute of Technology (KIT), Hermann-von-Helmholtz-Platz 1, D-76344 Eggenstein-Leopoldshafen, Germany; di.wang@kit.edu (D.W.); christian.kuebel@kit.edu (C.K.); 4Karlsruhe Nano Micro Facility, Karlsruhe Institute of Technology (KIT), Hermann-von-Helmholtz-Platz 1, D-76344 Eggenstein-Leopoldshafen, Germany; 5Department of Condensed Matter Physics, Charles University, Ke Karlovu 5, 121 16 Prague, Czech Republic; petr.machovec@matfyz.cuni.cz (P.M.); lukas.horak@matfyz.cuni.cz (L.H.); vaclav.holy@matfyz.cuni.cz (V.H.); 6Institute for Applied Materials, Karlsruhe Institute of Technology, Hermann-von-Helmholtz-Platz 1, D-76344 Eggenstein-Leopoldshafen, Germany; klaus.seemann@kit.edu (K.S.); sven.ulrich@kit.edu (S.U.); 7Laboratory for Applications of Synchrotron Radiation, Karlsruhe Institute of Technology, Kaiserstr. 12, D-76131 Karlsruhe, Germany

**Keywords:** heterostructures, temperature-dependent magnetic properties, pulsed laser deposition, interfacial quality, high-resolution transmission electron microscopy, high-resolution X-ray diffraction reciprocal space mapping, growth sequence

## Abstract

The structure and the chemical composition of individual layers as well as of interfaces belonging to the two heterostructures M1 (BaFe_12_O_19_/YbFeO_3_/YSZ) and M2 (YbFeO_3_/BaFe_12_O_19_/YSZ) grown by pulsed laser deposition on yttria-stabilized zirconia (YSZ) substrates are deeply characterized by using a combination of methods such as high-resolution X-ray diffraction, transmission electron microscopy (TEM), and atomic-resolution scanning TEM with energy-dispersive X-ray spectroscopy. The temperature-dependent magnetic properties demonstrate two distinct heterostructures with different coercivity, anisotropy fields, and first anisotropy constants, which are related to the defect concentrations within the individual layers and to the degree of intermixing at the interface. The heterostructure with the stacking order BaFe_12_O_19_/YbFeO_3_, i.e., M1, exhibits a distinctive interface without any chemical intermixture, while an Fe-rich crystalline phase is observed in M2 both in atomic-resolution EDX maps and in mass density profiles. Additionally, M1 shows high *c*-axis orientation, which induces a higher anisotropy constant K1 as well as a larger coercivity due to a high number of phase boundaries. Despite the existence of a canted antiferromagnetic/ferromagnetic combination (T < 140 K), both heterostructures M1 and M2 do not reveal any detectable exchange bias at T = 50 K. Additionally, compressive residual strain on the BaM layer is found to be suppressing the ferromagnetism, thus reducing the Curie temperature (Tc) in the case of M1. These findings suggest that M1 (BaFe_12_O_19_/YbFeO_3_/YSZ) is suitable for magnetic storage applications.

## 1. Introduction

Different devices were recently produced by combining barium ferrite BaFe_12_O_19_ (BaM) with ferroelectric materials to form multiferroic systems with magneto-electric coupling at the well-defined and characterized interface in terms of sharpness, atomic step, and chemical interdiffusion. It has covered thin-film heterostructures such as SrBa_2_Ta_2_O_9_/BaM [1,2], Ba_2_EuFeNb_4_O_15_/BaM [3,4], multilayers composed of BaM layers with Pb(Zr,Ti)O_3_ (PZT) [5] or (Ba,Sr)TiO_3_ (BST) [6,7,8] or BaTiO_3_ perovskite layers [9], and multiferroic composites [10,11,12]. BaM is a very attractive material due to the high anisotropy H_a_ and coercivity H_C_ fields [13,14], which are beneficial for obtaining performant film in perpendicular recording media. h-YbFeO_3_ belongs to the hexagonal rare earth ferrites h-(RFeO_3_) with R = Dy, where ferroelectricity and antiferromagnetism have been demonstrated at low temperatures [15]. Furthermore, room-temperature ferroelectricity has been confirmed for h-YbFeO_3_ grown on platinum Pt(111)/Al_2_O_3_(0001) [16] as well as for h-YbFeO_3_ grown by pulsed laser deposition on yttria-stabilized zirconia YSZ(111) [17,18]. The magnetic characterization of h-YbFeO_3_ was performed by measuring the polarization loops at different temperatures. Zero-field-cooled (ZFC) and field-cooled (FC) magnetic measurements demonstrated the existence of the magnetic transition Neel temperature of about 125 K [16,17,19]. Moreover, h-YbFeO_3_ has been demonstrated to possess a canted antiferromagnetic spin structure exhibiting a ferromagnetic moment [19]. There was an agreement about the existence of ferroelectricity and weak magnetic ordering in h-YbFeO_3_. Recently, there has been interest in building heterostructures based on h-YbFeO_3_. Xu et al. [20] designed a ferroelectric/dielectric bilayer structure with h-YbFeO_3_ as the improper ferroelectric (FE) and CoFe_2_O_4_ as the dielectric (DE) with the goal to tune the electrostatic energy of the (FE/DE) bilayer system and to study the electrostatic effect of the DE layer on the spontaneous polarization of the FE layer. Their atomic-resolution cross-sectional scanning transmission electron microscopy (STEM) high-angle annular dark-field (HAADF) imaging demonstrated a periodic arrangement with “two-up-one-down” and “two-down-one-up”, corresponding to the FE polarization [20,21,22]. The influence of the film thickness ratio of the FE to DE layers on the ferroelectric polarization has been discussed [21]. Heterojunction nanocomposites based on YbFeO_3_ were also successfully obtained by Tikhanova et al. [23] for application in photocatalysis, and a novel YbFeO_3_–BiFeO_3_ composite was synthesized by Chen et al. [24] for highly sensitive ppb-level acetone sensing at low temperatures. Zhang et al. [25] investigated the effect of the different interfaces on epitaxy and magnetism in h-RFeO_3_/Fe_3_O_4_/Al_2_O_3_ film (R = Lu, Yb) heterostructures. They demonstrated the enhancement of the magnetic remanence of the heterostructure h-RFeO_3_/Fe_3_O_4_/Al_2_O_3_ for temperatures below 50 K in comparison with the Fe_3_O_4_ layer. The degree of lattice misfit and the difference in the thermal expansion coefficient between individual layers play a relevant role in the growth of the heterostructure systems since they introduce microstructural strain and dislocations in the different layers. As a result, the film quality and the preferred growth orientation of YbFeO_3_ as well as its magneto-optical properties can be tailored by the choice of the substrate as it was demonstrated by Fu et al. [26]. This represents the most common approach used for altering the strain at the interface between the epitaxial film and the substrate and the resulting magnetic and electric properties [27,28]. Interfacial effects such as epitaxial strain [29], interface quality (i.e., defect, roughness, sharpness) [30,31,32], interdiffusion, and chemical intermixing [33,34] in multiferroic heterostructures have attracted tremendous attention for tuning and understanding the structure–property relationships as they allow one to control the magnetic interaction [35], spin ordering [36,37], and coupling across heterointerfaces, including ferromagnetic (FM)/antiferromagnetic (AFM) bilayer systems [30,32,33,34,35,36,37,38,39,40]. Several studies reported the occurrence of an exchange interaction based on the exchange bias (EB) at the interface of FM/AFM heterostructures. This phenomenon leads to a horizontal shift of the hysteresis loop with an enhanced coercive field H_C_ when the magnetization loop is recorded at temperatures below the Neel temperature T_N_ of the AFM after cooling the FM/AFM bilayers with an applied magnetic field. This was explained by pinning effects from uncompensated surface spins of the antiferromagnetic layer. However, in the case of a fully compensated AFM surface, where the AFM spins align themselves perpendicular to the FM spins to minimize the interfacial spin frustration, this resulted in a spin-flop coupling, which in return enhanced H_C_ without inducing an EB. The investigation by Vafaee et al. [32] into La_0.7_Sr_0.3_MnO_3_/BiFeO_3_ (LSMO/BFO) heterostructures revealed the absence of an EB coupling for multistacks with a sharp interface, whereas a sizable EB coupling was observed for (LSMO/BFO) heterostructures with rough and chemically mixed interfaces [41,42,43,44,45,46,47]. Furthermore, the structural misfit at the heterointerfaces in other types of multifunctional heterostructures (e.g., BaFe_12_O_19_/BaTiO_3_ [9], La_0.7_Sr_0.3_MnO_3_/BiFeO_3_ [32], LaMnO_3_/LaFeO_3_ [33], LaMnO_3_/LaNiO_3_ [48]) was tuned by the material combination as well as by the growth sequence with the goal to modify the exchange interaction [9,32,33,48].

Temperature-dependent magnetic properties such as coercivity and zero-field cooling (ZFC) and field cooling (FC) were studied by Vafaee et al. [32] for La_0.7_Sr_0.3_MnO_3_/BiFeO_3_ (LSMO/BFO) heterostructures. In their investigation, they demonstrated that ZFC and FC curves were influenced by the stacking order of the layers. In a similar way, Chen et al. [33] also revealed that the temperature-dependent magnetization curves were affected by the growth sequences of the layer for LaMnO_3_/LaFeO_3_ heterostructures due to the difference in the chemical and structural qualities of the interfaces.

In our study, the FM order of BaM was combined with a canted AFM (CAFM) and ferroelectric order in YbFeO_3_ (YbFO) for two different growth sequences on yttria-stabilized zirconia substrates with (111) orientation, i.e., YSZ(111).

The structures of M1(BaM/YbFO/YSZ) and M2(YbFO/BaM/YSZ) were characterized by a combination of X-ray reflectivity (XRR), high-resolution X-ray diffraction (HRXRD), high-resolution transmission microscopy (HRTEM) as well as atomically resolved scanning STEM combined with energy-dispersive X-ray spectroscopy (EDXS). The goal of our study was to analyze the effect of growth sequences on the structure of the heterostructures M1 and M2 as well as on their resulting magnetic properties which were dependent upon temperature, basing this on the quality of the corresponding interfaces.

## 2. Experimental Section

### 2.1. PLD Growth of Multiferroic Heterostructure Systems

Two heterostructure systems named M1(BaM/YbFO/YSZ) and M2(YbFO/BaM/YSZ) were deposited by pulsed laser deposition with two different stacking orders on YSZ(111) substrates. The corresponding schematic presentation of the systems is given in Figure 1h,k. We used BaM and YbFeO_3_ targets with 99.9% purity and YSZ(111) with a miscut of α_M_ ± 0.3° which was supplied from the company Surface, Germany (www.surfacenet.de). Prior to the growth, two pieces of YSZ(111) substrates which belonged to the same batch were cleaned with ethanol and isopropanol and annealed for 2 h under air in a furnace at 1200 °C. The individual BaM and YbFO layers in M1 and M2 were grown at *Tg* = 850 °C, with an oxygen pressure of *P* = 400 mbar, target–substrate distance Ts = 40 mm, and number of shots N = 10,000. Both M1 and M2 were grown with a laser frequency *F* of 1 Hz, an energy per pulse of 25 mJ (1.5 J/cm^2^), and a laser wavelength of 266 nm. The thicknesses for the BaM and YbFO layers which were grown under the above-mentioned conditions were approximatively *Th_BaM_* ≅ *Th_YbFO_* ≅ 60 nm. 

### 2.2. High-Resolution and X-ray Diffraction Reciprocal Space Mapping 

A Rigaku SmartLab diffractometer, equipped with a 9 kW rotating X-ray tube, was used to conduct theta/2theta X-ray diffraction measurements. High-resolution parallel-beam geometry was used, using Cu K_α1_ (*λ* = 1.540593 Å) radiation monochromatized with a Ge (220) 2-bounce monochromator. The measurements were taken with an incident slit width of 0.5 mm and two receiving slits, 0.5 mm and 1.0 mm wide, positioned in front of a scintillation detector.

Moreover, 2D-reciprocal space maps (2D-HRXRD) were recorded for the different BaM and YbFO reflections by using high-resolution diffraction at the NANO beamline at the KIT Light Source in Karlsruhe, Germany. All the 2D-HRXRD data of the symmetric and asymmetric reflections were measured using a Mythen linear detector positioned at the corresponding Bragg diffraction angles and by rocking the sample around the Bragg angle. All the X-ray measurements were performed at an energy *E* = 15 keV and a wavelength *λ* of 0.826 Å. Furthermore, azimuthal Phi(φ) scans were also measured by rotating the samples M1, M2, and BaM around the surface normal for the asymmetric reflections YbFO108 and BaM-1018.

The miscut was measured by using an Empyrean diffractometer. For the different azimuth angles in the range *ϕ* = [0–350°], the detector was positioned at the Bragg diffraction 2**Q_B_* of the symmetric YSZ111 reflection. We measured the deviation Δ*Q = (Q_max_ − Q_B_)* from the rocking curve scan which corresponded to the (111) crystal lattice planes of the YSZ substrate, where *Q_max_* is the angle of the diffraction peak maximum. The variation Δ*Q_XRD_* derived from XRD as a function of the azimuthal angle *ϕ* is shown by red open circles in Appendix A for M1 and M2, respectively.

For azimuthal angles *ϕ* equally space by 30° in the range of [0–350°], we measured the inclination of the sample surface in the reflectivity region where the detector was positioned at the exit angle corresponding to the specular reflections. The deviation of the incidence angle with respect to the exit angle gave Δ*Q_XRR_*, which is plotted in blue open circles in Appendix A for M1 and M2, respectively. The difference between the values of Δ*Q_XRD_(ϕ)* and Δ*Q_XRR_(ϕ)* for each azimuth value ϕ defined the angle between the film surface and the crystal lattice of the substrate in the given direction. These angles Δ*Q_XRD_(ϕ)* and Δ*Q_XRR_(ϕ)* were fitted by the function Δ*Q(ϕ) = A* × *sin(ϕ* − *ϕ_0_)*, and the amplitude A of the sine function was the miscut α of the substrate. The largest miscut *α_M_* = 0.67° was measured for the substrate of the sample M2, while for M1, it amounted to *α_M_* = 0.08°.

From the different 2D-HRXRD measurements, we derived the angular broadening of the diffraction profiles *FWHM_ang_* for the symmetric reflections of BaM (BaM006, BaM0010, BaM0014, and BaM0016) and of YbFO (YbFO004, YbFO006, and YbFO008) to determine the degree *α* of misorientation and therefore to evaluate the quality of the BaM and YbFO layers for the M1(BaM/YbFO/YSZ) and M2(YbFO/BaM/YSZ) heterostructures. The X-ray data analysis included the determination of the peak positions, which corresponded to the maximum intensities, to derive the out-of-plane and the in-plane lattice parameters for the BaM and YbFO crystals. In addition, we calculated the residual strain and the lattice misfit between the individual layers in the M1(BaM/YbFO/YSZ) and M2(YbFO/BaM/YSZ) heterostructures. All the derived values are summarized in Table 1.

The lattice misfits *f^M1^_BaM/YbFO_* (resp. *f^M2^_YbFO/BaM_*) corresponded to M1 (resp. M2) between the BaM/YbFO (resp. YbFO/BaM) bilayers. Additionally, the lattice misfits *f^M1^_YbFO/YSZ_* (resp. *f^M2^_BaM/YSZ_*) between the substrate and the film interface YbFO/YSZ (resp. BaM/YSZ) were calculated as follows (see Table 1b). Moreover, the misfit between the substrate and the film was calculated by using the 3 × d($11\bar{2}$)_YSZ_ formula of the substrate to the a-parameter of the layers. This is because of the coincidence 3-to-1 in the lattice site between the substrate and the film.
$$f_{BaM/YbFO}^{M1}\;\left[\%\right]=\frac{a_{M1\mathrm{BaM}}-a_{M1\mathrm{YbFO}}}{a_{M1\mathrm{YbFO}}}\times 100\;\mathrm{and}\;f_{YbFO/YSZ}^{M1}\left[\%\right]=\frac{a_{\mathrm{YbFO}}-3*d{\left(11\bar{2}\right)}_{\mathrm{YSZ}}}{3*d{\left(11\bar{2}\right)}_{\mathrm{YSZ}}}\times 100\;$$
$$f_{YbFO/BaM}^{M2}\;\left[\%\right]=\frac{a_{M2\mathrm{YbFO}}-a_{M2\mathrm{BaM}}}{a_{M2\mathrm{BaM}}}\times 100\;\mathrm{and}\;f_{BaM/YSZ}^{M2}\left[\%\right]=\frac{a_{M1\mathrm{BaM}}-3*d{\left(11\bar{2}\right)}_{\mathrm{YSZ}}}{3*d{\left(11\bar{2}\right)}_{\mathrm{YSZ}}}\times 100\;$$
where, the magnitudes *a_M1BaM_*, (resp. *a_M2BaM_*), *a_M1YbFO_* (resp. *a_M2YbFO_*), and $d{\left(11\bar{2}\right)}_{\mathrm{YSZ}}$ are the in-plane lattice parameters of the BaM, YbFO, and YSZ substrates, respectively. 

The in-plane residual strain (i.e., ε_M1YbFO//_ and ε_M2YbFO//_) of the YbFO layer and (i.e., ε_M1BaM//_ and ε_M2BaM//_) of the BaM layer was determined for the M1 and M2 heterostructures according to the following:$$\varepsilon_{M1\mathrm{BaM}//}=\frac{a_{M1BaM}-a_{BaM\;}^{FS}\;}{a_{BaM}^{FS}}\;\mathrm{and}\;\varepsilon_{M1\mathrm{YbFO}//}=\frac{a_{M1YbFO}-a_{YbFO\;}^{FS}\;}{a_{YbFO}^{FS}},$$
$$\varepsilon_{M2\mathrm{BaM}//}=\frac{a_{M2BaM}-a_{BaM\;}^{FS}\;}{a_{BaM}^{FS}}\;\mathrm{and}\;\varepsilon_{M2\mathrm{YbFO}//}=\frac{a_{M2YbFO}-a_{YbFO\;}^{FS}\;}{a_{YbFO}^{FS}}.$$

The out-of-plane residual strain (ε_M1YbFO⊥_, ε_M2YbFO⊥_) of the YbFO layer and (ε_M1BaM⊥_, ε_M2BaM⊥_) of the BaM layer were calculated using the following formulas:$$\varepsilon_{M1\mathrm{BaM}}=\frac{c_{M1BaM}-c_{BaM\;}^{FS}\;}{c_{BaM}^{FS}}\;\mathrm{and}\;\varepsilon_{M1\mathrm{YbFO}}=\frac{c_{M1YbFO}-c_{YbFO\;}^{FS}\;}{c_{YbFO}^{FS}},$$
$$\varepsilon_{M2\mathrm{BaM}}=\frac{c_{M2BaM}-c_{BaM\;}^{FS}\;}{c_{BaM}^{FS}}\;\mathrm{and}\;\varepsilon_{M1\mathrm{YbFO}}=\frac{c_{M2YbFO}-c_{YbFO\;}^{FS}\;}{c_{YbFO}^{FS}}.$$

$a_{BaM\;}^{FS}=5.892\;Å$, $c_{BaM\;}^{FS}=23.183\;Å$ (ICSD 201654, space group P63/mmc) and $a_{YbFO\;}^{FS}=5.9652\;Å$, $c_{YbFO\;}^{FS}=11.7020\;Å$ (ICSD 183152, space group P63 cm) correspond to the lattice parameters of the BaM and YbFO bulks in the free-standing (FS) states.

### 2.3. X-ray Reflectivity and Profile Density

The specular X-ray reflectivity (XRR) was measured by using a Rigaku Smartlab diffractometer. The data were measured using Cu-K_α1_ radiation and a parallel beam with an X-ray mirror. The data were evaluated by the well-known method for the XRR analysis of multilayers with rough interfaces developed by Parratt [49]. The used model for fitting the XRR curves of M1 and M2 is described as following: a rough YSZ substrate, a thin interlayer between the substrate and the first layer, the first layer with rough interface, and a rough top layer. The fit was carried out by using a self-written script based on the least square fitting algorithm. All the fitting parameters are summarized in Table 2a.

### 2.4. Transmission Electron Microscopy

For the TEM inspection of the M1(BaM/YbFO/YSZ) and M2(YbFO/BaM/YSZ) heterostructure systems, cross-sectional specimens were prepared by focused ion beam (FIB) milling and by using an FEI Helios G4 dual-beam microscope. Prior to the FIB preparation, a thin gold layer was sputtered on the sample surface in order to reduce the damage of the heterostructures caused by the Ga^+^-ion bombardment. Subsequently, the standard FIB preparation of TEM lamellae was carried out, where first a Pt protection layer was deposited on top of the samples. Then, coarse FIB milling was carried out at a primary ion energy of 5 keV. The lamellae were attached to Cu lift-out grids and finally polished by a Ga^+^-ion beam with a low energy of 1 keV in order to minimize Ga^+^ implantation and material amorphization. The TEM investigations of M1 and M2 were carried out on two different transmission electron microscopes. An image-corrected 300 kV FEI Titan 80–300 microscope was used for the conventional TEM analysis of the layer structure (e.g., layer thickness, crystal structure). The crystal structure and microchemistry of the interfacial regions and the structure and composition of the BaM and YbFO layers were investigated in detail by a combination of STEM imaging and energy-dispersive X-ray spectroscopy (EDXS) with a probe-corrected Thermo Fisher Themis 300 microscope which operated at 300 kV and was equipped with a Super-X EDX detector. STEM images were taken by using a high-angle annular dark-field (HAADF) detector to obtain atomic number contrast. X-ray maps were recorded in STEM mode with typical measurement times of about 10 to 20 min, where a possible drift of the sample was automatically corrected by the cross-correlation of corresponding reference images. Using the Velox software package (Thermo Fisher Scientific, https://assets.thermofisher.cn/TFS-Assets/MSD/Datasheets/velox-datasheet.pdf, accessed on 12 April 2024), raw-data X-ray maps were quantified using the thin-film approximation after Cliff–Lorimer [50] in order to obtain element concentration maps and quantitative EDXS line profiles. Moreover, crystal structure information was obtained by a two-dimensional fast Fourier transformation (FFT) of the selected areas in atomically resolved STEM images.

### 2.5. Vibrating Sample Magnetometry

The magnetic in-plane (IP) and out-of-plane (OOP) magnetization loops were measured by using a VersaLab (San Diego, CA, USA) vibrating sample magnetometer from Quantum design with a magnetic field up to 2 Tesla and a sweeping rate (SR) of 10 Oe/s. There were two setups used for the magnetization measurements. For standard (OOP) and (IP) measurements, the standard mode of VSM was used which allows measurements from 50 K to 400 K. We measured the Curie temperature *Tc* of the M1(BaM/YbFO/YSZ) and M2(YbFO/BaM/YSZ) systems by recording the moment in-plane (IP) moment M_//_ (H ⊥ to *c*-axis) during the heating process from 300 K up to 1000 K with a heating rate of 10 K/min in the VSM oven mode where the sample was wrapped by copper foil.

The measurements were performed for the M1(BaM/YbFO/YSZ), M2(YbFO/BaM/YSZ), and BaM (BaM/YSZ) systems at different temperatures T = 50, 70, 100, 120, 150, 200, 250, 300, and 400 K.

From the (IP) and (OOP) magnetization loops, the saturation magnetization Ms_//_(T), Ms_⊥_(T), the remanent magnetization Mr_//_(T), Mr_⊥_(T), the perpendicular and parallel squareness S_⊥_(T) = Mr_⊥_(T)/Ms_⊥_(T) and S_//_(T) = Mr_//_(T)/Ms_//_(T), the out-of-plane and in-plane coercivity fields Hc**_⊥_**(T) and Hc**_//_**(T), the coercivities ratio Rc(T)=Hc_//_/Hc_⊥_, and anisotropy field H_a_(T) were derived. For more reliability, the anisotropy fields H_a_(T) were determined using a more advanced approach where the effects of various factors on the shape of the magnetization reversal loop were considered as demonstrated by Zehner et al. [51]. The latter include the misalignment between the hard axis and direction of the applied magnetic field, the speed of sweeping as well as the domain structure and multiple magnetic phases. In this case, the average anisotropy fields H_a_(T) were obtained for M1(BaM/YbFO/YSZ) and a (IP) hysteretic magnetization M2(YbFO/BaM/YSZ) from the crossing of extrapolating the linear part of curve to the saturation and saturation magnetization level of the (OOP) curves. This was applied for all hysteresis loops measured at different temperatures T = 50, 70, 100, 120, 150, 200, 250, 300, and 400 K. Zero-field cooling (ZFC) and field cooling (FC) where the M1 and M2 samples were cooled without and with applied fields H, respectively, were recorded in the temperature range from T = 50 to 400 K with an applied field H = 2000 Oe (H is parallel to the *c*-axis). 

## 3. Results

### 3.1. Characterization of the Heterostructures M1 and M2

The diffraction patterns of M1(BaM/YbFO/YSZ) and M2(YbFO/BaM/YSZ), which simultaneously comprise the reflections of the BaM, YbFO, and the YSZ substrates, are compared in Figure 1a. Moreover, 2D-HRXRD maps were also recorded for the BaM reflections BaM006, BaM008, BaM0012, and BaM0016 as well as for the YbFO reflections YbFO002, YbFO004, and YbFO008, but only the 2D-HRXRD data of YSZ222, YbFO008, and BaM0016 are presented in order to investigate the crystallographic orientation of the different layers with respect to each other in the out-of-plane direction (Figure 1f,g).

The presence of high reflection orders, namely BaM0016 and YbFO008, justifies the *c*-axis orientation for both samples, i.e., M1 and M2. The XRD pattern of sample M2 shows a higher background/higher diffuse scattering intensity than that of the M1 sample. Apart from the diffuse scattering, the lower intensity of the BaM0016 and YbFO008 reflections of the M2 sample also shows the poor crystal quality compared to M1. However, the quality of the individual layers cannot be distinguished due to the overlapping of reflection intensities. From 2D-HRXRD maps of symmetric and asymmetric reflections, we determined the lattice parameters of the BaM and YbFO layers in both heterostructures M1 and M2. We summarize them in Table 1. We found that the out-of-plane lattice parameter *c_M_*_1*BaM*_ = 23.25 Å of the BaM layer in M1 is lower than that of the BaM layer in M2 (*c_M_*_2*BaM*_ = 23.2942 Å) as it could be demonstrated by the diffraction peak shift of M2 towards lower diffraction angles (see Figure 1a). Furthermore, the *c_M_*_1*YbFO*_ = 11.693 Å of the YbFO layer in M1 is slightly lower than the *c_M_*_2*YbFO*_ = 11.715 Å of the YbFO layer of M2. This difference in the lattice parameters may be due to the variation in the chemical composition of the individual BaM and YbFO layers in the heterostructures M1 and M2, despite the similar employed PLD growth conditions.

Due to the change in the stacking order of BaM and YbFO in M1 and M2, the BaM layer interfaces with YbFO in M1 and with YSZ in M2, the misfit varies from *f^M^*^1^*_BaM/YbFO_* [%] = −3.56% in M1 to *f^M^*^2^*_BaM/YSZ_* [%] = −5.30% in M2. This leads to different in-plane lattice parameters (a_M1BaM_ = 5.8151 ± 0.001 Å, a_M2BaM_ =5.967 ± 0.001 Å) and therefore in-plane residual strain which changes from compressive *ε_M_*_1*BaM//*_ = −1.3% to tensile *ε_M_*_2*BaM//*_ = 1.3% (see Table 1c). The misfit at the interface BaM/YbFO in M1 is *f^M^*^1^*_BaM YbFO_* = −3.56%, while in M2, it is *f^M^*^2^*_YbFO/BaM_* = 0.35%. Due to the difference in the above-mentioned misfit values between M1 and M2 at the BaM interfaces, the in-plane residual strain changes from compressive to tensile, but the absolute value does not conspicuously alter. In turn, we do not expect a strong variation in the Curie temperature *Tc* between M1 and M2.

The YbFO layer interfaces with YSZ(111) in M1 and with BaM in M2, where the misfit varies from *f^M^*^1^*_YbFO/YSZ_* [%] = −4.42% in M1 to *f^M^*^2^*_YbFO/BaM_* [%] = 0.35% demonstrating a change from compressive to tensile. This has slightly affected the in-plane lattice parameters which vary from a_M1YbFO_ = 6.0221 ± 0.001 Å to a_M2YbFO_ = 5.9881 ± 0.001 Å and therefore decreases the in-plane residual strain from ε_M1YbFO//_ = 0.96% to ε_M2YbFO//_ = 0.39% (see Table 1b,c).

Figure 1b,c show the azimuthal Phi(φ) scans along the surface normal of the BaM-1018 and YbFO108 reflections for both samples M1(BaM/YbFO/YSZ) and M2(YbFO/BaM/YSZ), respectively. Both layers show a 6-fold symmetry for both samples, indicated by the presence of reflections (each 60°). In Figure 1b, from the reflection intensities, it can be seen that the YbFO layer is 30°, rotated with respect to the BaM layer along the surface normal for sample M1, whereas in M2 (Figure 1c), the BaM and YbFO layers are well aligned with respect to each other. Therefore, the YbFO layer is rotated with respect to the underlayer in in-plane orientation. For the M1 and M2 heterostructures, the in-plane rotation relationship of the individual layers is visualized in Figure 1d,e by corresponding crystal structure models simulated along the [0001] direction.

The diffraction spots of BaM0016 and YbFO008 in Figure 1f are well aligned with respect to the YSZ222 diffraction spot along the q_ang_ direction for the M1 system, where the white dashed line crosses through the peak positions of the diffraction spots. This indicates that the *c*-axis of grown layers BaM and YbFO of M1 are well aligned with respect to the surface normal of the substrate YSZ(111). In the case of the system M2, BaM0016 and YbFeO008 diffraction spots show a clear horizontal shift in the *q_ang_* direction with respect to the YSZ222 diffraction spot (see white dashed line in Figure 1g). This indicates the formation of an inclination angle *β* = 0.1° for M2(YbFO/BaM/YSZ) between the *c*-axes of the BaM and YbFO layers with respect to the normal of the YSZ(111) crystal planes. A more reliable determination of the true misalignment of the YbFO and BaM bilayers with respect to the YSZ(111) lattice planes normal would require comparing RSMs recorded for different azimuthal angles and fit sinusoidal functions similar to the miscut case.

In the case of M2, the HRTEM images of Appendix A demonstrate the existence of steps at the substrate interface with an atomic resolution which is due to a miscut of the YSZ(111) in accordance with α_M_ = 0.67° measured by HRXRD. This can explain the origin of the recorded misalignment for M2(YbFO/BaM/YSZ).

In order to investigate the quality of the individual layers in the heterostructures M1(BaM/YbFO/YSZ) and M2(YbFO/BaM/YSZ), we compare the angular diffraction profiles of the BaM0014 and Yb006 reflections for M1 and M2 in Figure 2a,b, respectively. We found that the angular diffraction profile of the (0014) BaM reflection is broader for M1(BaM/YbFO/YSZ), which indicates that the defect density is probably higher in the BaM layer of the M1(BaM/YbFO/YSZ) heterostructure. We applied the mosaic model to the BaM and YbFO layers where the individual layers are composed of mosaic blocks misoriented due to defect formation. The variation in the *FWHM_ang_* broadening determined for the BaM and YbFO reflections is plotted as a function of the reflection order *00l* in Figure 2c,d, respectively. By applying the Williamson–Hall approach [52] for the *FWHM_ang_* versus *00l* plot, we derived the degree of misorientation of the BaM mosaic blocks from the slope. We found that *α_BaM_M_*_1_ = 0.74° is higher than *α_BaM_M_*_2_ = 0.41°, which confirms a higher defect density in the BaM layer of M1 in comparison to the BaM layer of M2. The cross-section TEM images of M1 and M2 in Figure 2e,f indicate a higher number of defect boundaries which are illustrated by yellow dashed curves for the BaM layers. The lateral sizes of BaM mosaic blocks vary in the range from 29 to 154 nm for M1 and in the range between 46 and 169 nm in the case of M2. The X-ray diffraction analysis in combination with TEM characterization enabled us to conclude that the BaM layer in M1 contains more defect boundaries than in the case of M2. However, a similar approach applied to the YbFO layers demonstrated similarities in the quality of the YbFO layer for both heterostructures despite the degree of misorientation being slightly different (*α_YbFO_M_*_1_ = 0.64° and *α_YbFO_M_*_2_ = 0.53°). Figure 3a,h show HRTEM micrographs of the M1 and M2 multiferroic systems, which include the different orientations of the different individual layers such as YSZ, YbFO, and BaM. For both M1 and M2 bilayer systems, BaM and YbFO were grown with the [001] direction parallel to the [111] direction of YSZ. A high-resolution STEM HAADF image of the YbFO/YSZ(111) interface in the M1 heterostructure is given in Appendix A where an interlayer with a thickness of *Th* = 2.5 nm and a different atomic arrangement can be seen that in addition exhibits a different image contrast than in the YbFO layer. Furthermore, there were no detectable atomic steps at the YbFO/YSZ(111) interface.

The YbFO layer in the M2(YbFO/BaM/YSZ) heterostructure shows a different interface with the BaM layer. The high-resolution STEM HAADF imaging of the YbFO/BaM interface (Appendix A) demonstrates the existence of well-resolved atomic steps at the YbFO(AFM)/BaM(FM) interface. It should be emphasized that the lattice misfit is reduced from *f^M^*^1^_YbFO/YSZ_ = −4.42% in M1 to *f^M^*^2^_YbFO/BaM_ = 0.35% in M2. The diffractograms in Figure 3c,e correspond to the YbFO regions marked with dashed green and orange squares in Figure 3a,h, respectively. Different diffraction spots are visible for M1 and M2 due to the different in-plane crystallographic orientation of the YbFO layers. We found that the in-plane orientation of the YbFO layer changes from [2-1-10] in M1 to [10-10] in M2. This indicates that YbFO grown on BaM(001) in M2 is rotated by 30° (±60°) around the [001] direction compared to the YbFO layer grown on YSZ(111) in M1 (cf. Figure 3c,e). 

In a similar way, we compared the crystal structure and orientation of the BaM layers in M1 and M2 based on the diffractograms in Figure 3b,d. It turns out that the in-plane orientation of the BaM layer is not affected by the growth sequence. 

However, the sharpness and the quality of the interface of BaM to YbFO in M1 and to YSZ in M2 are different. The TEM micrographs of M1 indicate a sharp interface between BaM and YbFO because of the low value of the lattice misfit *f^M^*^1^*_BaM/YbFO_* which is about −3.56%, while the BaM in the M2 system forms an interlayer when it grows directly on the YSZ(111) substrate. The nature of the interlayer will be discussed in detail in the following chapter.

The measured XRR curves of M1(BaM/YbFO/YSZ) and M2(YbFO/BaM/YSZ) together with the fits are given in Figure 4a. The variation in the mass densities *ρ* with penetration depth is plotted for M1 (black solid line) and M2 (red solid line) in Figure 4b, where the thickness *Th* = 0 nm corresponds to the surface of the heterostructures. In Figure 4b, we assume that the mass densities of the substrate YSZ as well as of the YbFO and BaM layers are constant as they are limited by short dashed vertical lines. Furthermore, the mass densities of the targets BaFe_12_O_19_, YbFeO_3_, and YSZ(111) are given by *ρ_BaFe_*_12*O*19_ = 5.296 g/cm^3^, *ρ _YbFeO_*_3_ = 6.8 g/cm^3^, and ρ_YSZ_ = 5.92 g/cm^3^, respectively, and are drawn by the horizontal gray dashed lines in Figure 4b. The transition regions which correspond to the interfaces with the YSZ(111) substrate (i.e., BaM/YSZ for M2 and YbFO/YSZ for M1) and between the bilayers (BaM/YbFO for M1 and YbFO/BaM for M2) are highlighted by green and gray rectangles to be investigated with higher magnification for the thickness x-axis scale in Figure 4c. The latter was devoted to clearly visualize the changes at the interface and to measure the dimensions of the transition regions at the mentioned interfaces in the heterostructures where the mass density varies with the thickness *Th*. The extension of the transition regions at the interfaces was estimated from the mass density profiles for M1 and M2 and are summarized in Table 2. For M1, the transition regions *R^M^*^1^*_Tr_*_1_ and *R^M^*^1^*_Tr_*_2_ (respectively *R^M^*^2^*_Tr_*_1_ and *R^M^*^2^*_Tr_*_2_ for M2) correspond to the interface at the substrate and between the individual layers. 

The interface near the YSZ(111) substrate for M1 *R^M1^_Tr1_* = 2.81 ± 0.1 nm is smaller than the *R^M2^_Tr1_* = 3.95 ± 0.1 nm of M2 as illustrated by green boxes in the right panel of Figure 4c. This indicates that the interface with the substrate is sharper in the case of M1. However, for both heterostructures M1 and M2, the mass density *ρ* of the layer which is deposited on YSZ(111) (YbFO for M1, BaM for M2) deviates from the one of the used target. This can be due to interdiffusion phenomena of substrate atoms into the grown layer. The chemical composition variation across the heterostructures is displayed in Figure 5 and Figure 6.

The left panel of Figure 4c compares the behavior of the transition regions *R^M1^_Tr2_* and *R^M2^_Tr2_* of the interfaces BaM/YbFO for M1 and YbFO/BaM for M2. We found that *R^M2^_Tr2_* = 10.3 ± 0.1 nm is larger than *R^M1^_Tr2_* = 8.2 ± 0.1 nm. This suggests the formation of a more distinguished interface BaM/YbFO for M1. Additionally, the mass density of YbFO slightly deviates from the one of the sputtered target YbFeO_3_. For M2, the BaM layer fits well with the stoichiometry of the BaFe_12_O_19_ target as demonstrated from the mass density profile which merges with *ρ_BaM_*.

In Figure 5a,j, high-resolution STEM HAADF images are given of the YbFO layer grown on YSZ(111) for M1(BaM/YbFO/YSZ) and BaM directly deposited on YSZ(111) for M2(YbFO/BaM/YSZ), where, in the rectangular regions marked by white boxes, combined STEM/EDXS analyses were performed. The corresponding STEM HAADF images and quantified X-ray maps for the elements Ba, Fe, Yb, Zr, and O are shown in Figure 5d–h for M1 and Figure 5m–r for M2. The obtained element-specific line profiles of M1 and M2 are given in Figure 5i,r. Figure 5i displays the profiles of Ba, Fe, Yb, Zr, and O for the heterostructure M1 in a region near the YbFO/YSZ interface, limited by the white box and crossing through the interface which is illustrated by the dashed magenta lines in Figure 5a. The latter contains a disturbed structure where a different atomic arrangement appears as a mixture between the pure YbFO layer and the pure YSZ substrate. Here, from the STEM HAADF image in Figure 5d and from the EDXS maps (Figure 5e–h) and the corresponding elemental profiles, we can distinguish four characteristic ranges *R^M1^_YSZ_*, *R^M1^_int1_*, *R^M1^_int2_*, and *R^M1^_YbFO_*. The region *R^M1^_YSZ_* = *Th* < 2.75 nm represents the pure substrate YSZ(111), while the region *R^M1^_YbFO_* = *Th* > 4.6 nm corresponds to a pure YbFO layer of M1 with corresponding slight concentration oscillations of Yb, Fe, and O. The transition region comprises *R^M1^_int1_* = 2.75 nm < *Th* < 4.6 nm, where the interdiffusion of Zr into the YbFO layer took place as measured by the simultaneous increase in Yb and Fe and decrease in Zr. Another region, *R^M1^_int2_* = 4.6 nm < *Th* < 6.5 nm, defines the deposited YbFeO film in M1 near the interface with residual Zr atoms from the substrate, which are not detected at *Th* = 6.5 nm away from the YSZ substrate. Therefore, the element concentration profile in *R^M1^_int1_* has the same behavior as in *R^M1^_YSZ_* with a progressive decrease in the Zr concentration, while *R^M1^_int2_* displays similar elements with a residual concentration of Zr.

In summary, the interlayer corresponds to a transition region with a thickness of about *Th* ≅ 4 nm *R^M1^_tr_* = *R^M1^_int1_* + *R^M1^_int2_* = 2.75 < *Th* < 6.5 nm. The evaluation of the mass density profiles for M1 from the XRR fitting gives a transition region between the mass densities of YSZ and YbFO with a thickness of *Th* = 2.3 nm, which mostly reflects the size of *R^M1^_int1_* = 2.75 nm < *Th* < 4.6 nm determined from the Zr and Yb elemental profiles.

Figure 5b,c depict the diffractograms of the YbFO and YSZ(111) substrates indicating the growth of a highly *c*-axis-oriented YbFO layer, despite the presence of the interdiffusion region.

From the STEM HAADF image contrast visible in Figure 5j,m, it can be clearly seen that the interfacial region between BaM and YSZ indicated by white dashed lines appears to be relatively diffuse. For M2, four defined regions *R^M1^_YSZ_*, *R^M2^_int1_*, *R^M2^_int2_*, and *R^M2^_BaM_* are found in the corresponding Ba, Fe, Yb, Zr, and O element profiles. The transition region *R^M2^_tr_* in the heterostructure M2 corresponds to *R^M2^_int1_* + *R^M2^_int2_* where *R^M2^_int1_* = 1.25 nm < *Th* < 2.9 nm shows an intermixing between the elements of the substrate (i.e., Zr) and those of the BaM layer (i.e., Ba, Fe). Furthermore, in the second part of the transition region *R^M2^_int2_* = 2.9 nm < *Th* < 4.8 nm, the Zr and O concentrations χ [at%] decrease, while the concentrations of Fe and Ba increase. This behavior suggests that the chemical composition of the interlayer regions *R^M2^_int1_* and *R^M2^_int2_* do not follow either the structure of the substrate or the one of the BaM top layer. This results in the formation of the transition region *R^M2^_tr_* with a thickness of about *Th* ≅ 3.55 nm between the YSZ substrate and the BaM layer in the system M2. Furthermore, the regions *R^M2^_YSZ_* and *R^M2^_BaM_* correspond to the pure substrate and the BaM layer in the M2 heterostructure, respectively. The high-resolution STEM image in Appendix A depicts different regions at the BaM/YSZ interface for M2, where the transition region *R^M2^_tr_* with a disturbed crystal structure and an accordingly diffuse image contrast extends over a thickness of *Th* = 4.6 to 6.7 nm across the interface. This finding is in accordance with the size of the transition region of the mass density profile measured by XRR at the BaM/YSZ for M2 and estimated to be about *Th* = 4.3 nm (see Figure 4c). Furthermore, M2 also contains few regions where the BaM/YSZ interface does exhibit a chemical intermixing, as shown in Appendix A.

The study of the interface chemistry near to the YSZ(111) substrate in both heterostructures M1 and M2 demonstrates the formation of a region with chemical intermixing over a thickness of about *Th* = 3.5 to 4 nm, which comprises the elements of the substrate and YbFO in M1, resp. BaM in M2.

Figure 5k,l show the diffractograms of the BaM and the interlayer *R^M2^_int2_*. It was not possible to record the inclination angle *β* = 0.1°, which is visible in the 2D-HRXRD reciprocal space map of Figure 1g. Furthermore, the diffractogram of Figure 5l which corresponds to *R^M2^_int2_* indicates the existence of crystalline phases, and the elemental profile reveals that these phases are rich in Fe and contain Ba as an impurity (see Figure 5r).

Figure 6a–g show the structure and chemical composition of the YbFO/BaM interfacial region in M2. For this purpose, we recorded large-scale X-ray maps (not shown here) corresponding to the STEM HAADF image in Figure 6a, and the derived element profiles are presented in Figure 6g. It should be noted that because of the large field of view of the scanned area in Figure 6a, the image resolution is relatively poor, and hence, single atomic columns are not resolved. However, we also provided atomically resolved STEM HAADF images and EDXS maps as shown in Figure 6b–f, which correspond to the region marked with a white box in Figure 6a. Generally, from the Ba, Fe, Yb, and O element profiles (see Figure 6g), we can distinguish three important regions, *R^M2^_BaM_*, *R^M2^_int3_*, and *R^M2^_YbFO_*, which are also indicated in the STEM image of Figure 6a. In Figure 6g, the regions *R^M2^_BaM_* and *R^M2^_YbFO_* are defined as regions of constant Ba and Yb concentrations χ, respectively, while *R^M2^_int3_* represents the transition region around the YbFO/BaM interface in M2, where a chemical intermixing is detected over an estimated extension across the interface of *R^M2^_int3_* = 9.11 ± 0.05 nm (cf. Table 2b). In addition, Figure 6g contains as an inset element profiles taken with atomic resolution that partially detect the interdiffusion of Ba atoms into the YbFO layer. The region *R^M2^_int3_* = 9.11 ± 0.05 nm measured from the EDXS profiles was found to be comparable with *R^M2^_Tr2_* = 10.3 ± 0.05 nm measured in the mass density profiles *ρ* derived from the XRR fitting procedure in the right panel of Figure 4c. Furthermore, the high-resolution STEM HAADF image of Appendix A shows an atomic step at the interface YbFO/BaM for M2. The structure and chemical composition of the M1 heterostructure were also explored at atomic resolution in a region near the BaM/YbFO interface for comparison with M2. Figure 6h depicts an atomically resolved STEM HAADF image of M1(BaM/YbFO/YSZ) in a region near the YbFO/BaM interface. From the region marked with an orange rectangle, X-ray maps were recorded and subsequently quantified (see Figure 6i–m). The resulting element concentration profiles of Ba, Yb, Fe, and O are compared in Figure 6n. Here, three characteristic regions, *R^M1^_YbFO_*, *R^M1^_BaM_*, and *R^M1^_int3_*, can be defined. *R^M1^_int3_* = 2.5 ± 0.05 nm = 4.5 < *Th* < 7 nm corresponds to the chemical transition region at the YbFO/BaM interface. The latter is found to be significantly reduced in comparison with *R^M2^_int3_* = 9.11 ± 0.05 nm and does not reveal any chemical intermixture between BaM and YbFO of the M1 bilayers as it can be deduced from the element profiles of Figure 6n. It should be mentioned that the transition region *R^M2^_Tr2_* = 8.2 ± 0.1 nm as determined from mass density (cf. Figure 4c) is relatively larger than *R^M2^_int3_* = 2.5 ± 0.05 nm measured from the element profiles (see Table 2b). From our EDXS investigation, we conclude that the degree of Yb/Ba intermixing depends on the stacking sequence. In fact, a relatively abrupt BaM/YbFO interface was detected in M1, while the YbFO/BaM interface in M2 appeared to be more chemically intermixed. Moreover, independently of the growth sequence, an atomic step at the YbFO/BaM and BaM/YbFO interfaces was revealed in M2 and M1, respectively, by atomically resolved STEM imaging, as depicted in Appendix A for both heterostructures, which could induce a magnetic frustration as reported by Chen et al. [53]. Regarding the crystal structure of the BaM and YbFO layers, the number of defect boundaries and the degree of misorientation of the BaM mosaic blocks were found to be affected by the stacking order of the layers (see Figure 2).

Appendix A show atomically resolved STEM HAADF images of the YbFO layers for the M1 and M2 heterostructures in areas of 12 nm × 12 nm. The diffractograms obtained by FFT analysis of the selected red and yellow square regions within the YbFO layers of M1 and M2 are shown in Appendix A, respectively. The diffractograms confirm that the growth sequence does not affect the ferroelectric metaphase of YbFeO_3_, which belongs to space group P63cm. Appendix A illustrate the atomic arrangement of the YbFO layer with representatively colored spheres. Green and yellow spheres represent Yb atoms, appearing with bright contrast in STEM HAADF images (green spheres displaced upwards and yellow spheres displaced downwards), while brown spheres represent Fe atoms with a darker HAADF contrast. As a result, and independently of the stacking order of the YbFO layer, ferroelectric domains were imaged for M1 and M2. In these domains, a non-centrosymmetric shift of Yb atoms generates charge polarization. Specifically, two Yb atoms are shifted vertically upwards, while one is shifted downwards which results in a ferroelectric polarization in the upward direction. Furthermore, Appendix A reveals a stacking fault (SF), which had formed by the absence of Yb atoms and the successive stacking of two Fe layers. This effect was only detected in a specific imaged region.

### 3.2. Effect of Growth Sequence on the Magnetic Properties

The out-of-plane (OOP) hysteresis loops recorded for the temperature range T = [50–400 K] for M1 and M2 are compared in Figure 7a, while the in-plane (IP) hysteresis loops are displayed in Figure 7b. The latter shows a higher IP area for the hysteresis loops in M1. Moreover, the IP coercivities and the remenance of M1 are higher than in the case of M2. Similarly, for the whole temperature range, we found that the OOP area, which corresponds to the dissipation energy, is large in the case of M1. One can deduce that the growth sequence changes the hard magnet M1(BaM/YbFO/YSZ) into a soft one M2(YbFO/BaM/YSZ) with a smaller area of the hysteresis loop (i.e., smaller hysteresis loss) and a lower coercivity field Hc_⊥_. Furthermore, independently of the stacking sequence, the OOP hysteresis loops display single magnetic hysteresis curves rather than the step-like hysteresis loop for the whole measured temperature range T = [50–400 K]. This is in accordance with the abrupt interface BaM/YbFO in M1(BaM/YbFO/YSZ) (Figure 6n) and the slightly intermixed interface YbFO/BaM in the case of M2(YbFO/BaM/YSZ) (Figure 6g). For better clarity, in Figure 7c, the OOP of the heterostructures M1(BaM/YbFO/YSZ) and M2(YbFO/BaM/YSZ) are compared for T = 50 and 300 K. In order to investigate the existence of interfacial magnetic coupling (EB) below the Neel temperature of YbFeO_3_ (T_N_ ≅ 125 K) [16,17,19] for M1(BaM/YbFO/YSZ) and M2(YbFO/BaM/YSZ), field-cooled hysteresis loops were measured after cooling to 50 K below the Neel temperature of the YbFO layer (i.e., T^YbFO^_N_ ≅ 140 K) under a field of +2 T and −2 T. The results given in Figure 7d do not show any detectable exchange bias (EB) for the M1 and M2 heterostructures at T = 50 K where the systems M1 and M2 correspond to (FM/CAFM) and (CAFM/FM), respectively. We did not consider to perform the EB measurement since the Neel temperature of YbFO was T^YbFO^_N_ ≅ 140 K, and the systems M1 and M2 underwent a transition to (FM/PM) and (PM/FM), respectively, where the EB was not relevant. 

The EDXS profiles of M2(YbFO/BaM/YSZ) demonstrate a chemical intermixture between Yb and Ba cations over an intermixed region of *R^M2^_int3_* = 8.7 ± 0.05 nm at the YbFO/BaM interface (Figure 6g, Table 2).

The chemical profile played a significant role in the magnetic coupling across the interface for other FM/AFM M1(BaM/YbFO/YSZ) and AFM/FM M2(YbFO/BaM/YSZ) heterostructures. It was found by Chen et al. [33] that a strong chemical intermixture between Mn^3+^ and Fe^3+^ generated an EB in the magnetization loops for LaMnO_3_/LaFeO_3_ due to uncompensated spins at the surface of the antiferromagnetic layer. Despite the presence of the intermixed region between the Yb and Ba atoms in the case of M2(YbFO/BaM/YSZ) at the interface YbFO/BaM, it seems that the spins of the YbFO layer at the interface were completely compensated. They energetically favored the spin flopping configuration which did not induce the EB in Figure 7d as has been recorded in the M1 and M2 heterostructures. Furthermore, several studies [32,33,34] confirmed the absence of an EB in the case of chemically abrupt interfaces similar to the one revealed in the case of M1(BaM/YbFO/YSZ). It is worthwhile emphasizing that YbFO is known by its canted antiferromagnetic spin orientations. This will strongly influence the spin ordering in the vicinity of the interface and remarkably contribute to the spin-flop phase as reported by Jensen et al. [37].

In order to understand and to compare the temperature dependence of magnetic properties for M1(BaM/YbFO/YSZ) and M2(YbFO/BaM/YSZ), the values of Ms_⊥_(T), Hc**_⊥_**(T), and H_a_(T) are derived from Figure 7 and plotted with the temperature in Figure 8. The analysis procedure of the (IP) and (OOP) curves is explained above in Section 3.2.

The Curie temperature T^FS^c of the BaM single crystal in the free-standing state was previously determined by Shirk et al. at Tc = 740 K [54]. In our case, the stacking sequence changed the in-plane residual strain in the BaM layer from compressive ε_M1BaM//_ = −1.3% in M1 to tensile in M2 ε_M2BaM//_ = 1.3%, while the out-of-plane residual strain increased from ε_M1BaM⊥_ = 0.29% to ε_M2BaM⊥_ = 0.48% (see Table 1c). The Tc was determined for M1 and M2 from the derivatives minimum dM/dT of the temperature–moment dependence M_//_(T) curve (see Figure 8a). We found T^M1^c = 723.5 ± 5 K and T^M2^c = 714.7 ± 5 K for the BaM layer in M1 and M2 which was lower than Tc = 740 K for BaM in the free-standing state. The lowering of the Tc of the strained film in comparison to the bulk has been also reported by Gan et al. in the case of epitaxial SrRuO_3_ films [28].

It is well known that the anisotropy constant K1(T) is related to the anisotropy field H_a_(T) and to the Ms_⊥_(T) and can be estimated from Equation (1) as follows [54,55,56,57,58]:(1)$$K1\left(T\right)=\frac{Ms_{⊥}\;\left(T\right)\times Ha\;\left(T\right)}{2}$$

Here, H_a_(T) was obtained from the intersection obtained by extrapolating the linear part of a (IP) hysteretic magnetization curve to saturation, while the saturation magnetization level Ms_⊥_(T) was determined from the (OOP) curves recorded at different temperatures T between 50 K and 400 K. As a result, the temperature dependences of Ms_⊥_(T) and K1(T) are given in Figure 8b,c. The decrease in M_s⊥_(T) by increasing T for the M1(BaM/YbFO/YSZ) and M2(YbFO/BaM/YSZ) heterostructures is due to the breaking of the spin ordering. Furthermore, M_s⊥_ (T) is lower in M2 than in M1, and this can be explained by an intermixed range R^M2^_int3_ = 8.7 ± 0.05 nm.

The anisotropy constant K1(T) also defines the magneto-crystalline anisotropy energy (MAE) which decreases with T due to the weakening of the spin orbital coupling (SOC) because of the thermal fluctuation. However, the values of K1(T) are found to be higher for M1(BaM/YbFO/YSZ) (see Figure 8c). This indicates that the SOC is influenced by the stacking sequence and is strong in the case of M1(BaM/YbFO/YSZ) which is more suitable for magnetic storage devices. The high values of K1(T) are in accordance with high *c*-axis orientation as revealed for M1(BaM/YbFO/YSZ) in contrast to M2(YbFO/BaM/YSZ), where the *c*-axis orientation of the heterostructures is off-axis with an inclination angle *β* = 0.1° with respect to the normal of the YSZ(111) lattice planes (see Figure 1f,g).

In order to compare the temperature-dependent saturation magnetization Ms_⊥_(T) and first-order anisotropy constant K1(T) between M1(BaM/YbFO/YSZ) and M2(YbFO/BaM/YSZ), both Ms_⊥_(T) and K1(T) were fitted by the modified Bloch’s law [57] as a power law, which is given by the following equations Equations (2) and (3) [56]. It has been demonstrated that the well-known Bloch’s T^3/2^ law, which is derived from the spin-wave theory of the thermal excitation of magnons, cannot fully describe Ms(T) for complex magnetic structures such as the hexaferrite [57,59].
(2)$$M_{s⊥}\;\left(T\right)=M_{s⊥}\;\left(0\right)\times \left[1-{\left(\frac{T}{T_{c}}\right)}^{P}\right]$$
(3)$$K1\;\left(T\right)=K1\;\left(0\right)\times \left[1-{\left(\frac{T}{T_{c}}\right)}^{P}\right]$$

(Ms_⊥_(0), P) and (K1(0), P) are the fitting parameters derived from the best fit of Ms_⊥_ (T) and K1(T), respectively, by using the non-linear square fit algorithms with chi-square values R = 0.999, which are shown in Figure 8b,c. The exponent *P* depends on the dimensionality, predominant spin, and crystal structure. In the case of homogeneous ferromagnetic materials, we apply *P* = 3/2 in Bloch’s law. It should be noted that T^M1^c = 723.5 ± 5 K and T^M2^c = 730.6 ± 5 K are used as fixed parameters as they were previously derived from Figure 8a.

The temperature dependence of the first anisotropy constant K1 and saturation magnetization Ms also obeyed the power law with the exponents P = 1.58 ± 0.04 and P = 1.56 ± 0.11, respectively, for silica-coated BaFe_12_O_19_ nanoparticles [56]. In the case of M1 and M2 heterostructures, Ms⊥(T) and K1(T) deviated from Bloch’s T^3/2^ law, which was applicable for example, to BaFe_12_O_19_ nanoparticles. The exponents derived from Ms⊥(T) were P^M1^_Ms⊥_= 1.277 ± 0.040 and P^M2^_Ms⊥_=1.253 ± 0.037 which were comparable for M1 and M2 in the range of uncertainty. On the other hand, the exponents derived from K1(T) were different from those derived from Ms⊥ (T). In fact, for M1, the exponent was P^M1^_K1_=1.432 ± 0.060, while for M2, it was P^M2^_K1_ = 1.861 ± 0.093 (see Table 3). As result, the difference between P^M1^_Ms⊥_ and P^M1^_K1_ for M1 was found to be smaller than the recorded the difference between the P^M2^_Ms⊥_ and P^M2^_K1_ values for M2. This can be related to the better homogeneity of M1 in comparison with M2, which exhibited a region of the chemical intermixing *R^M2^_int3_* = 8.7 ± 0.05 nm at the BaM/YbFO interface, affecting the resulting magnetic properties of the heterostructure M2. Similar phenomena were revealed by Garcia et al. who reported significant higher values for the exponents *P* in the case of yttrium iron garnets doped with Zn, Ni, and Co due to the atomic disorder and magnetic frustrations [59].

In order to explain the mechanism controlling the enhancement of the coercivity field and to understand the influence of the growth sequence on the temperature dependence Hc**_⊥_**(T), we use the following relation which is based on the micromagnetic model [58,60,61,62,63]:(4)$$\frac{H_{c\;⊥}\left(T\right)}{Ms_{⊥}\;\left(T\right)}=\alpha \times \frac{H_{a}\;\left(T\right)}{Ms_{⊥}\;\left(T\right)}-N_{eff},$$
where $N_{eff}$ is the demagnetization factor resulting from grain surface and volume charges, and α corresponds to a microstructural parameter which is also called the structural reduction factor [58]. The latter α = *α*_1_ × *α*_2_ is defined by Kronmüller et al. [58] as a product of *α*_1_ which is the pinning or the nucleation factor, and α_2_ is related to the misalignment factor of the grains with respect to the *c*-axis.

The plot $\frac{H_{c\;⊥}\left(T\right)}{Ms_{⊥}\;\left(T\right)}$ versus $\frac{H_{a}\;\left(T\right)}{Ms_{⊥}\;\left(T\right)}$ in Figure 8d enables us to determine the structural reduction factors *α^M1^* = 0.116 and *α*^M2^ = 0.04 from the slopes which are lower than *α* = 0.5, and this indicates that the mechanism which controls the coercivity is mixed. This includes the nucleation and the domain wall [58,60]. Since the misalignment with respect to *c*-axis is *β* = 0.1° for M2 and *β* = 0.01° for M1, as determined from 2D-HRXRD (Figure 1f,g), the factor α^M2^_2_ is probably higher than *α*^M1^_2_. In this case, the high value of *α^M1^* = 0.116 with respect to *α*^M2^ = 0.04 can be interpreted by the high number of pinning and nucleation centers which are located at mosaic boundaries which are increased in the case of the M1 heterostructure (see Figure 2e). This in turn explains the increase in the coercivity Hc^M1^⊥(T) values for M1 in comparison with M2, which contains fewer mosaic boundaries and therefore fewer pinning centers (Figure 8e). Moreover, the demagnetization factors N^M1^_eff_ = 0.072 and N^M2^_eff_ = −0.003, which were derived from the intercept of the y-axis, are relatively small (see Table 3). This can be interpreted by the reduced number of nonmagnetic phases in both heterostructures as demonstrated by Kronmüller et al. [58]. Nevertheless, this demagnetization factor N^M1^_eff_ is higher than in M2, indicating a higher defect density in the BaM layer of M1 despite the selectivity of the BaM/YbFO interface. The demagnetization factor N_eff_ was demonstrated to decrease with the grain misalignment and to be inversely proportional to the lateral size of the mosaic blocks [61]. This corresponds to the case of M2 where the misalignment β^M2^ = 0.1 and β^M1^ = 0.01 (Figure 1h,k) and the lateral size of the mosaic blocks is larger than in M1 (Figure 2e,f).

Appendix A presents the initial magnetization curves (IMC) of the M1 and M2 heterostructures, which were measured at T = 300 K to investigate domain wall motion and to determine the susceptibility of these systems. The inset in Appendix A provides a magnified view corresponding to magnetic fields below 5000 Oe. The curves of M1 and M2 exhibit a two-slope behavior, suggesting the potential existence of two distinct domain wall motion mechanisms. The first slopes are defined by the dashed lines as the tangent of the curves for the magnetic field up to approximately 1000 Oe. The latter are related with the predominant mechanism of domain wall pinning. For M1 and M2, beyond the field of 1000 Oe, the slope of the IMC changes and becomes steeper compared to the initial slope (<1000 Oe) as clearly shown in the inset of Appendix A. The critical coercivity *H_crit_* is defined as the field which corresponds to the change in the slopes in the IMC of M1 and M2. For the applied magnetic field below 1000 Oe, there is high material resistance which hinders the domain wall motion. On the contrary, beyond the magnetic field of 1000 Oe there is an increase in the curve slopes which indicates an overcoming of the resistance and enhancement of the domain wall motion mechanism. Furthermore, comparing the two first slopes of M1 and M2, we deduce that the slope of M1 is smaller than M2 which means the domain wall pinning effect is strong in M1. To determine accurately *H_crit_*, the second and third derivatives of the IMC (^d2^M/dH^2^ and d^3^M/dH^3^) were analyzed, as shown in Appendix A for M1 and Appendix A for M2. *H_crit_* is identified at the point where the third derivative equals zero. This marks the change in the slope of the IMC. For M1, H^M1^_crit_ = 1217 Oe, while for M2, H^M2^_crit_ = 913 Oe. This finding indicates that the depinning of domains in M1 occurs at a higher external field H^M1^_crit_ = 1217 Oe, which supports the hypothesis that a higher concentration of defects in a material requires a stronger field for domain wall depinning [64].

Figure 8e shows the temperature dependence of the (OOP) coercivity field Hc**_⊥_**(T) (bottom panel) and anisotropy field H_a_(T) (top panel) in the temperature range T = [50–400 K]. For both heterostructures M1 and M2, Hc**_⊥_**(T) and H_a_ (T) increase with T; however, the values of Hc**_⊥_**(T) and H_a_(T) are higher in the case of M1(BaM/YbFO/YSZ) with H^M1^c_⊥_ = [990 (50 K)–1296 Oe (400 K)] due to the probably high number of mosaic boundaries in the BaM layer as shown by TEM in Figure 2e,f. This in turn plays a role in domain wall pinning and therefore enhances Hc**_⊥_**(T) and consequently H_a_(T). The temperature dependence Hc**_⊥_**(T) and H_a_(T) of the BaM hexaferrite was investigated in earlier studies [54,55,57]. The increase in Hc**_⊥_**(T) and H_a_(T) reached a maximum beyond T = 400 K and below 500 K, followed by a decay toward the Curie temperature Tc = 740 K for the BaM single crystals [54]. In Figure 8f, zero-field cooling (ZFC^M1^_M⊥_, ZFC^M2^_M⊥_) and field cooling (FC^M1^_M⊥_, FC^M2^_M⊥_) curves are plotted for M1 and M2 as a function of temperature in the range T = 50–400 K with an applied field of H = 0 and H = 2000 Oe, respectively, where H is parallel to the *c*-axis. It should be reminded that H_app_ = 2000 Oe is higher than the coercivity of M1 H^M1^c_⊥_ = 990 (50 K)–1296 Oe (400 K) and M2 with H^M2^c_⊥_ = 407 (50 K)–448 Oe (400 K) (see Figure 8e). The high anisotropy constant K^M1^1 recorded for M1 lowered the moment ZFC^M1^_M⊥_ at T = 50 K in comparison with ZFC^M2^_M⊥_. Moreover, for both heterostructures M1 and M2, a thermal irreversibility (FC_M⊥_ > ZFC_M⊥_) was recorded due the magnetic anisotropy [65], but the degree of bifurcation (ZFC_M⊥_ − FC_M⊥_) was higher in the case of M2. This can be interpreted as the influence of the magnetic frustration resulting from the intermixed YbFO/BaM interface.

## 4. Conclusions and Discussion

On two different heterostructures M1 and M2 differing in the stacking of BaM layers as FM and YbFO as canted AFM and FE layers, a detailed study on the structure and the magnetic properties was carried by using complementary analysis methods (i.e., 2D-HRXRD, high-resolution TEM and STEM, atomically resolved EDXS). The difference in the interfacial strain, which was induced by the variation in the growth sequence in M1 and M2, was demonstrated to affect the crystal quality and the chemical composition of the individual BaM and YbFO layers as well as the selectivity and the degree of the chemical homogeneity at the interfaces.

By means of atomic-resolution EDXS, we explored the chemical composition of the interfaces with a YSZ substrate as well as at the BaM/YbFO interface in M1 and YbFO/BaM in M2. Interdiffusion phenomena were detected for M1 and M2 with different chemical compositions at the YSZ substrate interfaces, where a rather Fe-rich crystalline phase was formed at the BaM/YSZ interface in M2. The determined values of the misfit at the YSZ interface in M1 and M2 are comparable and compressive (i.e., *f^M^*^1^*_YbFO/YSZ_* = −4.42 [%], *f^M^*^2^*_BaM/YSZ_* = −5.53 [%]). The chemical composition of the specific layer that interfaces with the substrate was found to be different from the original stoichiometry of the target as demonstrated by the measurement of the mass densities.

While the interface between BaM and YbFO in M1 was sharp and without any chemical interdiffusion transition over a thickness of *R^M^*^1^*_int_*_3_ = 2.5 ± 0.05 nm, the YbFO/BaM interface in M2 revealed a chemical intermixing over an extension of *R^M^*^2^*_int_*_3_ = 8.7 ± 0.05 nm. This was most probably related to the change in the interfacial misfit from compressive *f^M^*^1^*_BaM/YbFO_* = −3.56 [%] to tensile *f^M^*^2^*_YbFO/BaM_* = 0.35 [%] strain. For both heterostructures M1 and M2, there was no detectable exchange bias, even in the case of the chemically intermixed interface for the M2 heterostructure.

It should be emphasized that below the Neel temperature of YbFO < T_N_ ≅ 140 K, the heterostructures M1 and M2 acted as FM/CAFM and CAFM/FM, respectively, while in the temperature range beyond T_N_ ≅ 140 K, M1 and M2 were close to FM/PM and PM/FM, respectively. Furthermore, the YbFO layers did not exhibit any variation in the metastable hexagonal p63cm ferroelectric ordering of Yb atoms. Even with varying stress and stacking orders, both systems exhibited the ferroelectric order, which is known as the non-centrosymmetric displacement of the rare earth element (also known as 2up/1down vice versa). The measured magnetic properties of the heterostructures were found to be strongly influenced by the ferromagnetic BaM layer due to the lower magnitude of the YbFO layer moment. The magnetic behavior of M1 and M2 was remarkably related to the crystal quality of the BaM layer which was crucially influenced by its interface properties, including in particular the residual strain, misfit, and chemical homogeneity of the layer and the interfacial region.

For the BaM layers, there was a slight crystallographic misalignment of the layer with respect to the substrate surface in the M2 system. The presence of an interlayer (such as YbFO for M1) was sufficient to suppress the misalignment that originated from the miscut of the substrate crystal resulting from its production. HRXRD, HRTEM, and atomically resolved STEM HAADF imaging confirmed that the BaM layer grown on YSZ had a lower defect concentration because of fewer mosaic boundaries despite the occurrence of an unwanted interlayer formation at the substrate interface.

The temperature dependences of magnetic properties such as Ms_⊥_, anisotropy constant K1, coercivity Hc_⊥_, and anisotropy H_a_ fields were compared between M1 and M2 with the goal to investigate the effect of the growth sequence. As a result, a highly *c*-axis-orientated BaM layer was obtained in the case of M1, with higher H_a_, Ms_⊥_, and K1 than for BaM in the heterostructure M2.

The extended intermixed region of M2 at the YbFO/BaM interface disturbed the homogeneity of the magnetic layers which induced variable deviations in the values of the exponents derived from Ms_⊥_(T) and K1(T) in the case of M2 because of the differences in anisotropy (and hence differences in homogeneity).

Furthermore, the intermixing of materials affects the ZFC-FC behavior, where film irregularities result in spin frustration, generating thermal irreversibility and disturbing the regular spin alignment, finally ultimately increasing the degree of bifurcation.

Additionally, it has been proven that tensile stress increases the T_C_ for BaM, while compressive strain suppresses ferromagnetism and reduces the T_C_. In conclusion, M1 with a distinct interface exhibits more homogeneous magnetic properties without an exchange bias phenomenon. On the other hand, M1 exhibits a larger coercivity, a higher density of defects (mosaic boundaries), a higher anisotropy constant, and a higher MAE. These characteristics suggest that M1 is better suited for magnetic storage applications.

## Figures and Tables

**Figure 1 nanomaterials-14-00711-f001:**
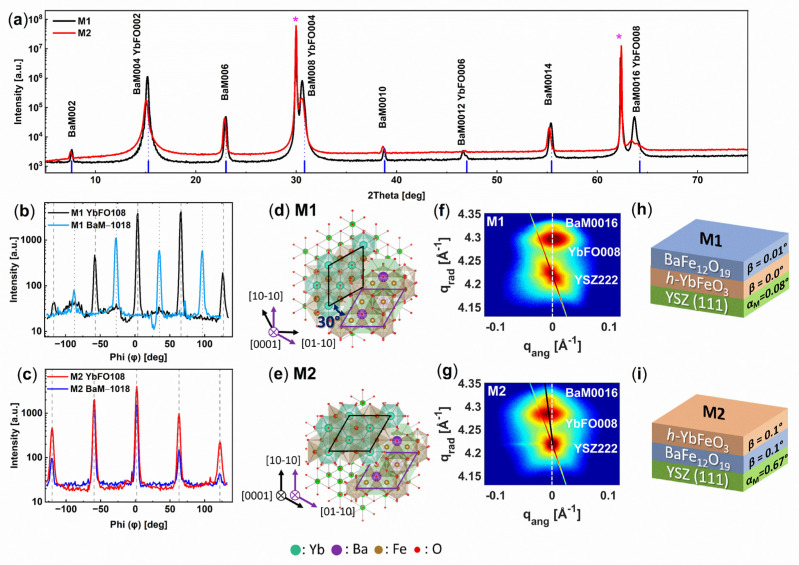
(**a**) Diffraction patterns of the heterostructures M1(BaM/YbFO/YSZ) (solid black curve) and M2(YbFO/BaM/YSZ) (solid red curve), substrate peaks are shown with (*) symbol. (**b**,**c**) Azimuthal scans of asymmetric reflections BaM(−1018) and YbFO(108) for M1 and M2, respectively. (**d**,**e**) Visualization of the in-plane rotation relationship of the atomic layers relating to the crystal structures as it is simulated along the [0001] direction for M1 and M2, respectively. (**f**,**g**) 2D-HRXRD reciprocal space maps containing the BaM(0016), YbFO(008), and YSZ(222) diffraction spots. (**h**,**i**) Schematic presentation of the heterostructures M1(BaM/YbFO/YSZ) and M2(YbFO/BaM/YSZ) with two different stacking sequences.

**Figure 2 nanomaterials-14-00711-f002:**
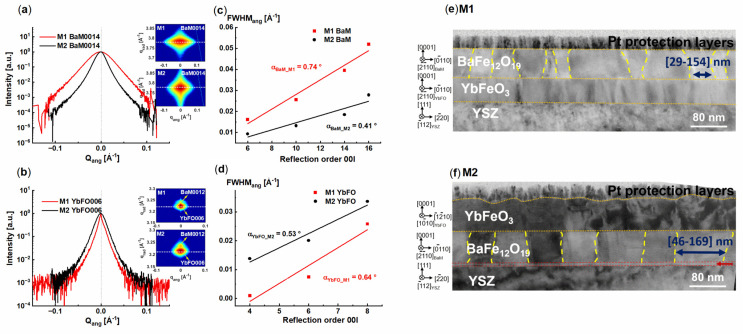
(**a**,**b**) Comparison between the M1(BaM/YbFO/YSZ) and M2(YbFO/BaM/YSZ) heterostructures by the angular diffraction profiles of the BaM0014 and YbFO006 reflections, respectively. (**c**,**d**) Comparison of M1 and M2 regarding the variation in the full width half maximum of the corresponding angular broadening *FWHM_ang_* with the reflection order *00l* for the BaM and YbFO layers, respectively. (**e**,**f**) TEM cross-section images of the M1 and M2 heterostructures, respectively, indicating the mosaic boundaries with yellow dashed lines.

**Figure 3 nanomaterials-14-00711-f003:**
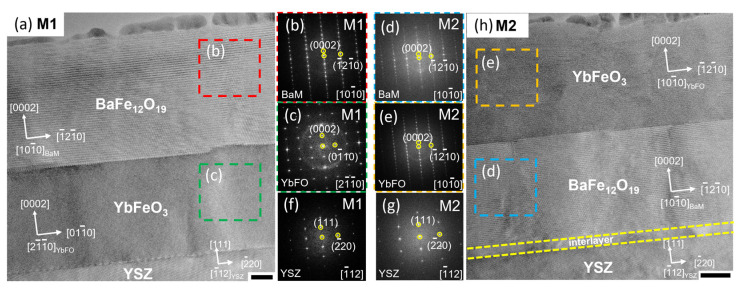
(**a**,**h**) HRTEM images of M1(BaM/YbFO/YSZ) and M2(YbFO/BaM/YSZ) in cross-section with the crystallographic orientation of the individual layers BaM, YbFO, and YSZ. (**b**,**c**,**f**) Diffractograms corresponding to M1(BaM/YbFO/YSZ) of the BaM layer drawn by a red box in (**a**) of the YbFO layer illustrated by a green box (**a**) and of the YSZ(111) substrate, respectively. (**d**,**e**,**g**) Diffractograms corresponding to M2(BaM/YbFO/YSZ) of the BaM layer drawn by a blue box in (**a**) of the YbFO layer illustrated by an orange box (**h**) and of the YSZ(111) substrate, respectively. Scale bars of (**a**,**h**) are in size of 20 nm.

**Figure 4 nanomaterials-14-00711-f004:**
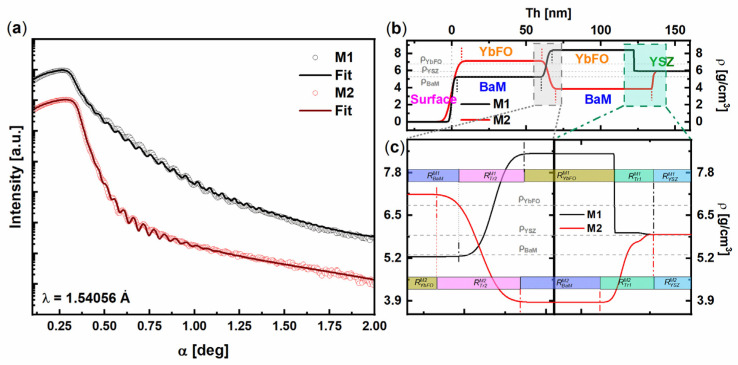
(**a**) Measured XRR curves (points) and the obtained fits (lines) for M1 (black) and M2 (red). The curves are vertically shifted for better visibility. (**b**) Density profiles obtained from the fits of M1 (black) and M2 (red). (**c**) Magnified sections of the interfaces of the density profiles marked with dashed squares in (**b**). The (**c**) left panel shows the interface between the layers and the (**c**) right panel shows the YSZ/first layer interface. The density profiles in (**c**) are shifted with respect to each other for a better comparison of the interfaces.

**Figure 5 nanomaterials-14-00711-f005:**
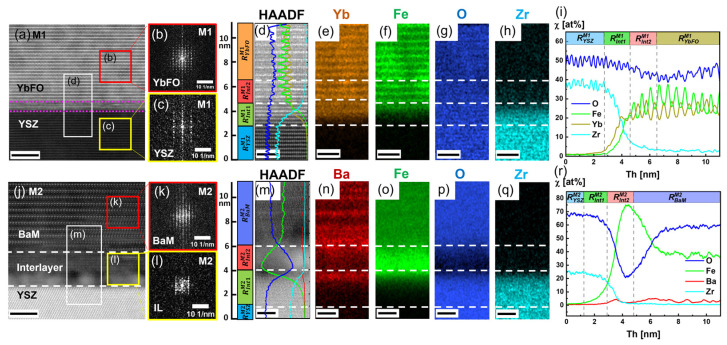
(**a**,**j**) High-resolution STEM HAADF images near the bilayer interface regions (YbFO/YSZ) and (BaM/YSZ) for the M1 and M2 heterostructures, respectively. (**b**,**c**) Diffractograms corresponding to red and yellow rectangles drawn in the BaM layer and YSZ substrate, respectively. (**k**,**l**) Diffractograms of the red and yellow rectangles drawn in the BaM layer and YSZ substrate, respectively. The regions marked with white rectangles are the regions selected for the corresponding X-ray maps given in (**d**–**h**) for M1 and in (**m**–**q**) for M2. (**i**,**r**) Comparison of the quantitative EDXS line profiles of the elements O, Fe, Ba, and Yb for M1 and M2, respectively. Scale bars of (**a**,**j**) are 5 nm and of (**d**–**h**) and (**m**–**q**) are 2 nm in size.

**Figure 6 nanomaterials-14-00711-f006:**
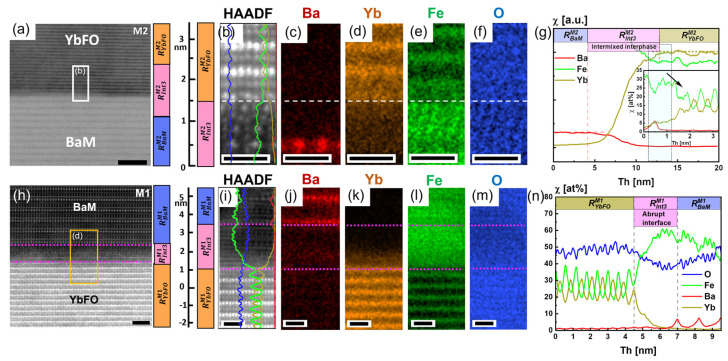
(**a**,**h**) High-resolution STEM HAADF images recorded from regions near the bilayer interface (YbFO/BaM) and (BaM/YbFO) for the M2 and M1 heterostructures, respectively. The regions marked with white and orange rectangles are the regions selected for corresponding X-ray maps given in (**c**–**f**) for M2 and in (**j**–**m**) for M1, (**g**,**n**) comparison of the quantitative EDXS line profiles of the elements Ba, Yb, for M2 and M1. The inset in (**g**) shows the magnified EDXS line profile illustrated by dashed square regions and black arrow. Scale bars of (**a**) are 5 nm, of (**h**) are 2 nm, and of (**b**–**f**) and (**i**–**m**) are 1 nm in size.

**Figure 7 nanomaterials-14-00711-f007:**
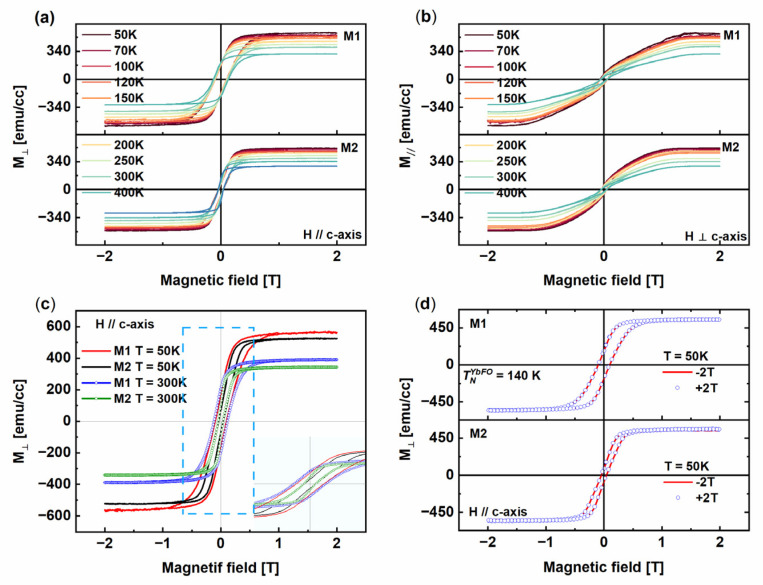
(**a**,**b**) Out-of-plane (OOP) and in-plane (IP) magnetization polarization curves for M1 (top panel) and M2 (bottom panel), respectively, recorded in the temperature range T = [50–400 K]. (**c**) Comparison between the OOP magnetization of M1 and M2 at T = 50 K and 300 K with the inset as magnification in the region marked with blue dashed square of H = [−1 T–1 T]. (**d**) OOP magnetization loop measured at 50 K and for applied magnetic field H = 2 T and −2 T for M1 (top panel) and M2 (bottom panel).

**Figure 8 nanomaterials-14-00711-f008:**
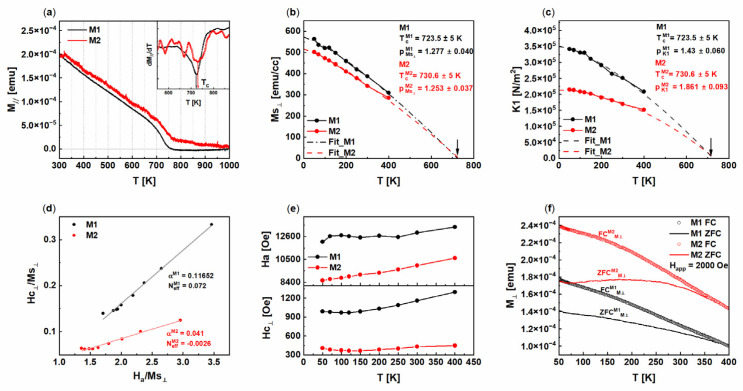
(**a**) Variation in the in-plane moment M_//_ versus temperature in the range T = [300–1000 K] for M1 and M2, and inset corresponds to the derivative dM/dT as function of temperature, the minima correspond to the Curie temperature T_C_. (**b**) Variation in the out-of-plane moment at the saturation Ms_⊥_ with the temperature for M1 (black full circle) and M2 (red full circle), and the fitting curves are the power law curves. (**c**) Temperature dependence of the fist anisotropy constant K1 for M1 (black full circle) and M2 (red full circle). (**d**) Variation in Hc_⊥_/Ms_⊥_ as function of the Ha/Ms_⊥_ (black full circle) and M2 (red full circle) dedicated to determine microstructural and demagnetizing factors *α* and *N_eff_*, respectively. (**e**) Anisotropy field H_a_ temperature dependence (top panel) for M1 and M2 and comparison between M1 and M2 of dependence of perpendicular coercivity Hc_⊥_ with temperature (bottom panel). (**f**) Zero-field cooling (ZFC) and field cooling (FC) recorded with applied field H = 2000 Oe for M1 (open circles) and M2 (solid lines).

**Table 1 nanomaterials-14-00711-t001:** (a) Summary of the lattice parameters corresponding to the different layers BaM, YbFO, and YSZ of the heterostructures M1 and M2. (b) Summary of the lattice misfit determined at different interfaces BaM/YbFO, YbFO/YSZ for M1 and YbFO/BaM, BaM/YSZ for M2. (c) Summary of the in-plane and out-of-plane residuals determined in the BaM and YbFO layers in the two different heterostructures M1 and M2.

(a)										
	**d(11-2)_YSZ_ [** **Å]**	**d(111)_YSZ_** **[Å]**	**a_M1YbFO_** **[Å]**	**a_M2YbFO_** **[Å]**	**c_M1YbFO_** **[Å]**	**c_M2YbFO_** **[Å]**	**a_M1BaM_** **[Å]**	**a_M2BaM_** **[Å]**	**c_M1BaM_ [Å]**	**c_M2BaM_ [Å]**
M1	2.100 ± 0.001	2.969 ± 0.001	6.0221 ± 0.001		11.693 ± 0.001		5.8151 ± 0.001		23.25 ± 0.001	
M2	2.100 ± 0.001	2.969 ± 0.001		5.9881 ± 0.001		11.7149 ± 0.001		5.967 ± 0.001		23.2942 ± 0.001
(b)										
			***f^M1^_YbFO/YSZ_* [%]**		***f^M1^_BaM/YbFO_* [%]**		***f^M2^_BaM/YSZ_* [%]**		***f^M2^_YbFO/BaM_* [%]**	
M1			−4.42 compressive		−3.56 compressive					
M2							−5.30 compressive		0.35 tensile	
(c)										
	**ε_M1YbFO//_** **[%]**	**ε_M1YbFO⊥_** **[%]**	**ε_M1BaM//_** **[%]**		**ε_M1BaM_** ** _⊥_ ** **[%]**	**ε_M2BaM//_** **[%]**	**ε_M2BaM_** ** _⊥_ ** **[%]**	**ε_M2YbFO//_** **[%]**	**ε_M2YbFO_** ** _⊥_ ** **[%]**	
M1	0.96 Tensile	−0.08 Compressive	−1.3 Compressive		0.29 Tensile					
M2						1.3 Tensile	0.48 Tensile	0.39 Tensile	0.11 Tensile	

**Table 2 nanomaterials-14-00711-t002:** (**a**) Summary of the fitting parameters such as mass densities ρ_BaM_ of BaM and ρ_YbFO_ of YbFO layer derived from the XRR curves of M1 and M2 and the corresponding density profiles across the heterostructures’ films thicknesses. (**b**) Summary of the characteristics of regions with their corresponding thicknesses of the EDX profiles of the two heterostructures as they are derived from Figures 5 and 6. (**c**) Summary of the concentration χ [at %] of the Yb, Fe, and Ba derived from the EDX profiles of Figures 5i and 6n for M1 and from EDX profiles of Figures 5r and 6g for M2.

(a)								
	**BaM thickness** **Th_BaM_ [nm]** **XRR**	**YbFO** **thickness** **Th_YbFO_ [nm]** **XRR**	**Mass density** **ρ_BaM_ [g/cm^3^]** **XRR**	**Mass density** **ρ_YbFO_ [g/cm^3^]** **XRR**	**Transition region** **YbFO/YSZ** ***R^M1^_Tr1_* [nm]** **XRR**	**Transition region** **BaM/YbFO** ***R^M1^_Tr2_* [nm]** **XRR**	**Transition region** **BaM/YSZ** ***R^M2^_Tr1_* [nm]** **XRR**	**Transition region** **YbFO/BaM** ***R^M2^_Tr2_* [nm]** **XRR**
M1	61.0 ± 0.2	58.8 ± 0.2	5.24 ± 0.05 *≅ ρ_BaFe12O19_* = 5.296 g/cm^3^	8.38 ± 0.05 *> ρ _YbFeO3_* = 6.8 g/cm^3^	2.81 ± 0.1	8.2 ± 0.1		
M2	70.0 ± 0.2	65.0 ± 0.2	3.86 ± 0.05 < *ρ_BaFe12O19_* = 5.296 g/cm^3^	7.14 ± 0.07 slightly > *ρ _YbFeO3_* = 6.8 g/cm^3^			3.95 ± 0.1	10.3 ± 0.1
(b)								
	**BaM thickness** **Th_BaM_ [nm]** **TEM**	**YbFO** **thickness** **Th_YbFO_ [nm]** **TEM**	**YbFO/YSZ** ***R^M1^_int1_* [nm]** **EDX profiles**	**YbFO/YSZ** ***R^M1^_int2_* [nm]** **EDX profiles**	**BaM/YbFO** ***R^M1^_int3_* [nm]** **EDX profiles**	**BaM/YSZ** ***R^M2^_int1_* [nm]** **EDX profiles**	**BaM/YSZ** ***R^M2^_int2_* [nm]** **EDX profiles**	**BaM/YbFO** ***R^M2^_int3_* [nm]** **EDX profiles**
M1	63 ± 0.2	53 ± 0.2	2.75 < *Th* < 4.6 1.85 ± 0.05 nm Interdiffusion of Zr atoms	4.6 < *Th* < 6.5 1.9 ± 0.05 nm Residual Zr atoms	4.5 < *Th* < 7 2.5 ± 0.05 nm Sharp interface without intermixing			
M2	67 ± 0.2	74 ± 0.2				1.25< *Th* < 2.9 1.65 ± 0.05 nm Interdiffusion of Zr atoms	2.9 < *Th* < 4.8 1.9 ± 0.05 nm Fe rich phase	4.2 < *Th* < 12.93 8.7 ± 0.05 nm Chemical intermixing
(c)								
	**BaM thickness** **Th_BaM_ [nm]** **TEM**		**YbFO thickness** **Th_YbFO_ [nm]** **TEM**	***R^M1^_YbFO_*.** **χ [at %]**		** *R^M1^_BaM_* ** **χ [at %]**	***R^M2^_YbFO_*.** **χ [at %]**	** *R^M2^_BaM_* ** **χ [at %]**
M1	63 ± 0.2		53 ± 0.2	χ_Yb_ (Max)= 30 ± 1 <χ_Yb_> = 23.57 <χ_Fe_> = 27.12 (see Figure 5i)		χ_BaM_ (Max) = 7 ± 0.5 <χ_Ba_> = 3.7 <χ_Fe_> = 54.03 (see Figure 6n)		
M2	67 ± 0.2		74 ± 0.2				χ_Yb_ (max) = 18 ± 0.5 Large scan <χ_Yb_ > = 19.85, <χ_Fe_> = 17.97 Atomic resolution <χ_Yb_ > = 13.89, <χ_Fe_ > = 18.53	χ_BaM_ (Max) = 6 ± 0.5 Large scan <χ_Ba_ > = 3.18, <χ_Fe_> = 36.77 Atomic resolution <χ_BaM_> = 3.39, <χ_Fe_> = 27.1

**Table 3 nanomaterials-14-00711-t003:** Summary of the characteristic magnetic parameters for M1 and M2 such as moment at saturation T = 0 K, Ms⊥(0) and exponents P, which were derived from the fitting of temperature dependence moment with the power law Ms⊥(0) × (1 − (T/Tc)^P^) and K1(0) × (1 − (T/Tc)^P^). Microstructural parameter *α* and demagnetization factor *Neff* were derived from Hc_⊥_/Ms_⊥_ as function of the Ha/Ms_⊥_.

	Moment at Saturation T = 0 K Ms⊥(0) [emu/cc]	Exponent P	Curie Temperature Tc [K]	Anisotropy Constant K1, T = 0 K1(0) [N/m^2^]	Exponent P	Microstructural Parameter *α*	Demagnetization Factor *Neff*
M1	Ms^M1^⊥(0) = 573.43 ± 5.97	P^M1^_Ms⊥_ = 1.277 ± 0.040	T^M1^c = 723.5 ± 5	K^M1^1(0) = 3.5E+6 ± 4.4E+3	P^M1^_K1_ = 1.432 ± 0.060	α^M1^ = 0.116	N^M1^_eff_ = 0.072
M2	Ms^M2^⊥(0) = 513.77 ± 5.68	P^M2^_Ms⊥_ = 1.253 ± 0.037	T^M2^c = 730.6 ± 5	K^M2^1(0) = 2.1E+6 ± 2.6E+3	P^M2^_K1_ = 1.861 ± 0.093	α^M2^ = 0.041	N^M2^_eff_ = −0.0026

## Data Availability

The data presented in this study are available on request from the corresponding author.

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
