# Peer review of "Dependence of the Structural and Magnetic Properties on the Growth Sequence in Heterostructures Designed by YbFeO3 and BaFe12O19"

_nanomaterials, 2024, doi:10.3390/nano14080711_

Round 1
Reviewer 1 Report
Comments and Suggestions for Authors
The authors, Sondes Bauer and colleagues, present findings on the structural and magnetic properties of BaFe12O19/YbFeO3 and YbFeO3/BaFe12O19 on a YSZ substrate. However, upon reviewing this manuscript, it becomes apparent that it lacks adequate preparation and organization. Regrettably, it is not yet in a state suitable for submission to a scientific journal.
For example, the first paragraph of the introduction is overly verbose and lacks coherence in its structure, imply listing all relevant references does not effectively introduce the manuscript.
The equation labels are presented in a confusing and unclear manner, making it difficult for readers to understand and reference them appropriately.
There are two 3.2 sections.
Line 650, the reference is missing
Line 544, what is the “specific”
These issues are merely the beginning of a myriad of mistakes present in the manuscript. Therefore, I strongly recommend that the authors thoroughly reorganize the paper to ensure it is presented in a coherent and proper manner.
Comments on the Quality of English Language
The English has to be improved.
Author Response
Dear Sir and Madame,
Please find here attached the revised version of our manuscript entitled “Dependence of the structural and magnetic properties on the growth sequence in heterostructures designed by YbFeO3 and BaFe12O19”. We appreciate the valuable and fruitful proposals of the referees. For easy tracking, we include all the recommended changes by the referees in blue color with the goal to improve the coherence and the structure of manuscript and to enhance the clarity of the discussions and the interpretations. Additionally, you will find detailed reply to each comment, indicating the corresponding line number of the added explanation and references basing on the comments of the referee. We are very thankful to the editor and to the referees for their critical reading of the manuscript and for their beneficial suggestions.
Yours sincerely,
Sondes Bauer

Reviewer 2 Report
Comments and Suggestions for Authors
The manuscript titled „Dependence of the structural and magnetic properties on the 2 growth sequence in heterostructures designed by YbFeO3 and BaFe12O19” shows interesting results worth publication and after minor revision can be published in Nanomaterial journal. Even thou the issues are numerous I do not consider them critical to preventing publication. Mostly concern clarification of sentences or editorial correction:
- Describing STEM HAADF authors discuss the intermixing of the BAM/YSZ interface and show it in Fig S2b, but Fig S2c also shows this interface without the intermixed region and is not mentioned in the text. What are the differences between those two images? Does the presence of these two images mean that there are two kinds of interface regions, ones with and seconds without intermixing?
- Could the authors specify in the 2.1 section the thicknesses of the layers.
- Could the authors specify why for calculations of the lattice misfits between the layer and the substrate a 3*d(11-2) was used (lines 176 and 177).
- Line 149 should describe both Qmax and 󠄀’spiral'B instead a 󠇯'spiral'󠇯max is given.
- The description of magnetic measurements (section 2.5) contains repeating information. It could be shortened. Also, the last sentence referring to Fig 8f should be removed.
- In line 363 authors refer to Fig S2e, but it should be S2d.
- The end of the sentence in lines 408-410 should be changed, figures cannot be analysed and discussed, they can show something.
- Authors few times include the same information about the source of spin frustration and possible induced exchange bias. I would suggest removing some of those sentences (i.e. sentences in lines 365-366 and 498-499 ) and leaving only the last one in line 518 adding information about possible exchange bias effect.
- The sentence in lines 522-523 is a repetition of information given a few lines above and can be removed
Comments about magnetic properties that I found more troublesome:
- Some discussion about IP hysteresis curves would be useful.
- In line 550 authors use the term “reduced hysteresis loop”. Could the authors specify what they have in mind?
- The calculation of K1 values seems problematic. The expression is valid for very idealistic cases of very well-defined easy and hard magnetic axes, which you do not have. It is well seen from small opened M(H) curves at low magnetic fields. Furthermore, the extraction of Ha could be done in different ways giving different values of K1. The one you used does not have to be valid for all temperatures, since, judging by the presented graph, the high-temperature IP M(H) curves could not be fully saturated (especially for M2). Therefore, I would suggest changing the word calculated (in line 596) to estimated. But this would have some consequences discussed below.
- The calculation of Bloch law should be modified for the K1(T) part. In Line 641 authors write that the exponents derived for K1(T) and Ms(T) are different. It is true but I cannot agree with the following text where the exponent PM1K1 is considered the correct one because it gives a value close to the Bloch exponent. As explained previously, from my point of view, the exponents obtained from Ms are more realistic than the ones from K1 dependencies. The PM1K1 could suggest better homogeneity of the M1 system than M2 because it gives a smaller deviation from PM1Ms value, and not because it is closer to P=3/2. The problem here is if the differences in magnetic anisotropy between, or PM1K1 and PM2K1, indeed are forced by the changes in homogeneity or are caused by the way of calculating the Ha values. Therefore I suggest modifying a little the text to be less categorical and avoid statements like confirms.
Also, the sentence in lines 606-609 should be moved to the end of the page and combined with the paragraph describing the effect.
- The description of ZFC/FC curves is not precise (also in the Experimental section). The authors write that they measure the ZFC with an applied field of H=0 Oe while FC with H=2000 Oe. But it is incorrect, ZFC should be measured in the same field as the FC part but after cooling the sample in zero field while the FC after cooling in the H=2000Oe field. The authors probably did that while the description is not precise.
- Another issue of the ZFC/FC is the conclusion that the splitting is due to the spin frustration. However, the problem is that authors have the ferromagnetic material. When the ZFC/FC is measured in a temperature range lower than the TC and the field lower than needed for saturation, the ZFC and FC will always split. You are just measuring the M(T) dependences for different magnetization states of the ferromagnet. Even the differences in splitting between M1 and M2 samples cannot say anything certain. To measure the ZFC, you probably demagnetize the samples at 400K but that could lead to different demagnetized states and different splitting. This part also has to be modified and suggest that these changes can be interpreted as the influence of magnetic frustration.
- In Conclusions in line 855 authors mention the phase transition of YbFO and the change in the heterostructure between FM/CAFM ->FM/PM (or CAFM/FM ->PM/FM) but this change in properties is not mentioned in the results section. It would be good to add such information in the results section when describing magnetic properties. Furthermore later in the text authors mention that the lack of the influence of AFM on the FM layer is because of the weakness of AFM, could the authors explain what kind of weakness you have in mind?
- In the paragraph in lines 877-882 authors state that the M2 shows deviation from Bloch law. However, both samples do not show T3/2 dependence. Both samples are heterostructure where ferromagnet is one part of the material, therefore they do not have to give exactly the 3/2 exponent. My point is that you should not concentrate on the one value of exponent that was close to the 3/2 but rather emphasise that both samples show similar exponents for Ms(T) dependences while the K1 shows different deviations from them suggesting differences in anisotropy (and hence differences in homogeneity).
- In line 885 authors use the term “regular magnetic properties”, what those it mean?
Editorial problems:
- The obligatory statements are not filled, like “Data Availability Statement”, Funding, Authors Contributions, and Supplementary Materials. Even the text for “Figures, tables and Schemes” citing rules is left (line 729).
- Hyperlink is missing, line 650.
- Tables have very poor quality, as well as graphs.
- In lines 172-174, the description of lattice misfits is confusing and probably missing some parts.
- Line 544 to be removed.
- The section for Figures and Tables should have the number 3.3.
Author Response

(The authors gave the same response as above.)

Reviewer 3 Report
Comments and Suggestions for Authors
In this work, two multilayers BaFeO3 (BaM) and YbFeO3 (YbFO) were epitaxial growth on YZO substrates. The detailed structures at different grown sequential, i. e. BaFeO3 on YbFeO3 and YbFeO3 on BaFeO3 were carefully measured together with their corresponding magnetic properties. It is interesting see the subtle difference in interface structure on differing in the stacking of BaM layers as FM and YbFO as canted AFM and FE layers, results in the structure and correlated the magnetic properties in great details. The difference in the interfacial strain, which was induced by the variation of the growth sequence, was demonstrated to affect the crystal quality and the chemical composition of the individual BaM and YbFO layers as well as the selectivity and the degree of the chemical homogeneity at the interfaces. Sequentially, different magnetic properties, such Hc, Ha, K1, Tc, TN, FC/ZFC curve changed based this interface difference. The paper is of interesting and with high quality of scientific understanding also. It should be accepted for publication.
However, some writing should be improved:
Line 126: A typo, 1.5 J/cm2. The “2” should be in superscript.
Line 147, 148: the “Qmax, @max, Delta -@, @B” are not defined well.
For the lattice parameter a, c. What is the superscript “FS” represented? Is it mean free-standing state? Please defined it.
Line 544: specific?
In magnetic session: the notation of Ms^(T) and 𝑀𝑠⊥(T) are confused. Are they the same one?
Line 650: [Error! Bookmark not defined.]
Line 866: “For BaM layers, there is a slight crystallographic misalignment of the layer with respect to the substrate surface in the M2 system. Even if the misalignment originates from the miscut of the substrate.” It seems the second sentence is not complete sentence, or these sentences should be combined with a “,” instead of “.”.
Author Response

(The authors gave the same response as above.)

Reviewer 4 Report
Comments and Suggestions for Authors
The authors have investigated the dependence of the structural and magnetic properties on the growth sequence in heterostructures designed by YbFeO3 and BaFe12O19 using the pulsed laser deposition on yttria-stabilized zirconia (YSZ) substrate. Overall, minor revision in the manuscript is required to clarify certain points before final submission in Nanomaterials. The comments on the present manuscript are given below.
1) I recommend adding more results of this study in the abstract.
2) I recommend in the introduction to explore in more detail how variations in temperature impact the coercivity, anisotropy fields, and first anisotropy constants of the heterostructures.
3) Please provide band alingments for the M1 and M2 heterostructures, which can be used to support the discussion on their magnetic behavior.
4) I believe these recent studies may help in this work and I recommend adding these references in this manuscript.
https://doi.org/10.1038/s41467-024-44728-y
https://doi.org/10.3390/nano11092197
https://doi.org/10.1021/acsami.0c09934
https://doi.org/10.3390/ma17040871
https://doi.org/10.1016/j.jmmm.2022.170304
Author Response

(The authors gave the same response as above.)

Round 2
Reviewer 1 Report
Comments and Suggestions for Authors
The authors revised the format and language of the manuscript, but the introduction still can be improved.
On the science part, the authors report the structural and magnetic properties of YbFeO3/BaFe12O19/YSZ and BaFe12O19/YbFeO3/YSZ, by using a combination of XRD, atomic resolution TEM, and magnetic measurements. In general, this is a systematic study and deserves to be published in Nanomaterials. Following are some minor revision suggestions before the acceptance.
1. The high-resolution TEM images of M2 specimen, how is the interlayer be determined? What is the origin of the formation of such an interlayer?
2. Cross-section TEM images of M1 and M2 in Figures 2e and 2f indicate a higher number of boundaries, what is this boundary? antiphase boundary or impurity boundary? Better to clarify this.
3. Some labels in HRTEM are too small for readers, even when I magnify the image, they are still blurry. For example Fig.5i, s, and Fig. 6g,n.
After the minor revision is done , the manuscript can be accepted.
Comments on the Quality of English Language
The language of some sentences needs to be revised, for example, the following appears on line 113-115:
Temperature-dependent magnetic properties such as coercivity and zero field cooling (ZFC) and field cooling (FC) were studied by Vafaee et al. [32] La0.7Sr0.3MnO3/BiFeO3 (LSMO/BFO) heterostructures and demonstrated to be influenced by the stacking order.
Author Response
Dear Sir and Madame,
Please find here attached the revised version 2 of our manuscript entitled “Dependence of the structural and magnetic properties on the growth sequence in heterostructures designed by YbFeO3 and BaFe12O19”. We appreciate the meaningful recommendations of the referees. For easy tracking, we include all the proposed changes by the referees in the second round by red color, being additionally highlighted in turquoise.
You will find detailed reply to each comment, indicating the corresponding line number of the added explanation and references basing on the comments of the referee. We are very grateful to the editor and to the referees for their attention and their careful reading of the manuscript as well as for their valuable proposals.
Yours sincerely,
Sondes Bauer

Reviewer 4 Report
Comments and Suggestions for Authors
The authors made the requested modifications, for this reason, I recommend the publication of the work
Author Response

(The authors gave the same response as above.)
